# Cal-QL: Calibrated Offline RL Pre-Training for Efficient Online Fine-Tuning

## Abstract

A compelling use case of offline reinforcement learning (RL) is to obtain an effective policy initialization from existing datasets, which allows efficient fine-tuning with limited amounts of active online interaction in the environment. Many existing offline RL methods tend to exhibit poor fine-tuning performance. On the contrary, while naïve online RL methods achieve compelling empirical performance, online methods suffer from a large sample complexity without a good policy initialization from the offline data. Our goal in this paper is to devise an approach for learning an effective offline initialization that also unlocks fast online fine-tuning capabilities. Our approach, calibrated Q-learning (Cal-QL) accomplishes this by learning a conservative value function initialization that underestimates the value of the learned policy from offline data, while also being calibrated, meaning that the learned value estimation still upper-bounds the ground-truth value of some other reference policy (e.g., the behavior policy). Both theoretically and empirically, we show that imposing these conditions speeds up online fine-tuning, and brings in benefits of the offline data. In practice, Cal-QL can be implemented on top of existing offline RL methods without any extra hyperparameter tuning. Empirically, Cal-QL outperforms state-of-the-art methods on a wide range of fine-tuning tasks from both state and visual observations, across several benchmarks.

## 1. Introduction

Modern machine learning successes across many domains follow a common recipe: first pre-training large and expressive neural network models on general-purpose, internet-scale data, followed by fine-tuning the pre-trained initialization on a limited amount of data for the task of interest (He et al., 2022; Devlin et al., 2018). How can we translate such

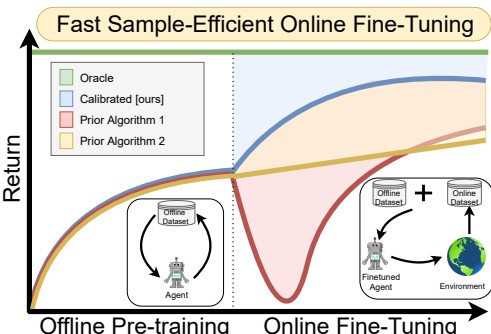

Figure 1. We study the problem of **online fine-tuning followed by offline RL pre-training**. Some prior offline RL methods tend to exhibit slow performance improvement in this setting (yellow), resulting in worse asymptotic performance, while others suffer from initial performance degradation (red), resulting in a high regret. We develop an approach that "*calibrates*" the learned value function estimates to enable the best of both worlds (blue).

a recipe to sequential decision-making? A natural way to instantiate this paradigm is to utilize offline reinforcement learning (RL) algorithms (Levine et al., 2020) for initializing value functions and policies from previously collected static datasets, followed by task-specific online fine-tuning that aims to improve this initialization with the smallest amount of active interaction. If successful, such a recipe might enable effective and generalizable online RL with significantly fewer samples than current RL methods that learn from scratch.

Many algorithms for offline RL have been applied to online fine-tuning. Empirical results across prior works suggest a counter-intuitive trend: policy initializations obtained from more effective offline RL methods tend to exhibit worse online fine-tuning performance (see Table 2 of Kostrikov et al. (2021b) & Figure 4 of Anonymous (2023)). On the other end, naïve online RL methods training from scratch (or RL from demonstrations (Vecerik et al., 2017), where the replay buffer of a standard online RL algorithm is seeded with the offline data) seem to improve online at a significantly faster rate. But these online methods require actively collecting data by rolling out policies in the environment from scratch, which inherits similar limitations of naïve online RL methods in problems where data collection is expensive or dangerous. Overall, these results suggest that it is difficult to devise an offline RL algorithm that both acquires a good initialization from prior data but also enables efficient fine-tuning.

[1]Anonymous Institution, Anonymous City, Anonymous Region, Anonymous Country. Correspondence to: Anonymous Author <anon.email@domain.com>.

Preliminary work. Under review by the International Conference on Machine Learning (ICML). Do not distribute.

How can we devise a method to learn an effective policy initialization that also improves during fine-tuning? Kumar et al. (2020) has shown that one can learn a good policy initialization by optimizing the policy against a *conservative* value function obtained using the offline dataset. But, we find in Section 4.1 that, conservatism alone is not sufficient for efficient online fine-tuning as these methods often tend to unlearn the policy initialization learned from offline data and "waste" samples collected via online interaction in re-covering this initialization, after which fine-tuning proceeds normally. Our analysis reveals that this is a result of the fact that value estimates produced via conservative methods can be arbitrarily smaller than the ground truth return of *any* valid policy. Such Q-value estimates that do not lie on a similar scale as the return of a valid policy in the MDP are problematic – once fine-tuning begins, actions executed in the environment that are actually worse than the policy learned from offline data will erroneously appear better if their ground-truth return value is larger than the learned con-servative value estimate. Subsequent policy improvement will begin to lose the initialization in favor of such a worse policy until the method recovers.

If we can also ensure that the conservative value estimate learned using the offline data is *calibrated*, meaning that the value estimate is on a similar scale as return values on the task, and larger than the ground-truth return of policies worse than the learned policy initialization, then we can avoid this issue. Of course, we cannot enforce such a condi-tion perfectly. Therefore, we choose to enforce the learned value estimates to be larger than the ground-truth values of only a *reference policy* whose value is known or can be estimated easily, such as the behavior policy. Even though this condition is not perfect, we show that it still leads to sample efficient online fine-tuning. Our practical method, **calibrated Q-learning (Cal-QL)**, learns conservative value functions using a conservative Q-learning (CQL) (Kumar et al., 2020) like training objective, while also explicitly forcing them to be calibrated against the behavior policy.

Our main contribution is an approach, Cal-QL, for acquir-ing an offline initialization that facilitates sample efficient online fine-tuning. Cal-QL aims to learn conservative value functions, that are also calibrated with respect to a refer-ence policy (e.g., the behavior policy in our experiments). Our analysis of Cal-QL shows that both theoretically and empirically, Cal-QL attains stronger guarantees on cumula-tive regret during fine-tuning. In practice, Cal-QL can be implemented over conservative Q-learning, a prior offline RL method, without any additional hyperparameters. We empirically evaluate Cal-QL across a range of benchmark tasks from Fu et al. (2020) and Nair et al. (2020a), including dexterous manipulation, navigation, and robotic manipula-tion tasks, and find that it matches or outperforms the best methods on all tasks, in some cases by 30-40%.

## 2. Related Work

Recent works in theory and practice both suggest that online RL methods typically require a large number of samples (Sil-ver et al., 2016; Berner et al., 2019; Vinyals et al., 2019; Ye et al., 2020; Kakade & Langford, 2002; Agarwal et al., 2019; Zhai et al., 2022; Gupta et al., 2022; Li et al., 2022) to learn from scratch. Instead, we can utilize offline data to accelerate online off-policy RL algorithms. Prior works instantiate this idea in a variety of ways: incorporating the offline data into the replay buffer of online RL (Schaal, 1996; Vecerik et al., 2017; Hester et al., 2018; Song et al., 2022), utilizing auxiliary behavioral cloning losses along-side policy-gradients (Rajeswaran et al., 2017; Kang et al., 2018; Zhu et al., 2018; 2019), or extracting a high-level skill space for downstream online RL (Gupta et al., 2019; Ajay et al., 2020). While these prior methods substantially improve the sample efficiency of running online RL from scratch, as we will also show in our results, they do not eliminate the need to actively roll out dangerous or poor policies for data collection.

To address this issue, a different line of work aims to first run offline RL for learning a good policy and value ini-tialization from the offline data, followed by online fine-tuning (Nair et al., 2020b; Kostrikov et al., 2021a; Lyu et al., 2022; Beeson & Montana, 2022; Wu et al., 2022; Lee et al., 2022; Mark et al.). These approaches typically employ existing offline RL methods based on policy constraints (Fu-jimoto et al., 2018a; Siegel et al., 2020; Guo et al., 2020; Ghasemipour et al., 2021; Kostrikov et al., 2021a) or pes-simism on the offline data for some training epochs, then continue training with the same method on a combination of offline and online data once fine-tuning begins (Nachum et al., 2019; Kidambi et al., 2020; Yu et al., 2020; Kumar et al., 2020; Buckman et al., 2020). Although pessimism is crucial for offline RL (Jin et al., 2021b; Cheng et al., 2022), using pessimism or constraints for fine-tuning (Nair et al., 2020b; Kostrikov et al., 2021a; Lyu et al., 2022; Beeson & Montana, 2022) slows down fine-tuning or leads to initial unlearning, as we will show in Section 4.1. In effect, these prior methods either fail to improve as fast as online RL or lose the initialization from offline RL. We aim to address this limitation by understanding some conditions on the of-fline initialization that enable fast fine-tuning and then turn these conditions into a fine-tuning method – Cal-QL.

Perhaps the most closely related to this work are methods that utilize a naïve pessimistic offline RL method for of-fline training, but incorporate optimism in fine-tuning (Lee et al., 2022; Mark et al.; Wu et al., 2022). In contrast, our method, Cal-QL aims to learn a better offline initialization that enjoys the benefits of offline RL pre-training but is more amenable to naïve fine-tuning. Our approach fine-tunes naïvely without ensembles (Lee et al., 2022) or exploration bonuses (Mark et al.), but attains good performance by learn-

ing an offline initialization that satisfies certain conditions.

## 3. Preliminaries and Background

The goal in RL is to learn the optimal policy for an MDP $\mathcal{M} = (\mathcal{S}, \mathcal{A}, P, r, \rho, \gamma)$. $\mathcal{S}, \mathcal{A}$ denote the state and action spaces. $P(s'|s, a)$ and $r(s, a)$ are the dynamics and reward functions. $\rho(s)$ denotes the initial state distribution. $\gamma \in (0, 1)$ denotes the discount factor. Formally, the goal is to learn a policy $\pi : \mathcal{S} \mapsto \mathcal{A}$ that maximizes cumulative discounted value function, denoted by $V^\pi(s) = \frac{1}{1-\gamma} \sum_t \mathbb{E}_{a_t \sim \pi(s_t)} [\gamma^t r(s_t, a_t)|s_0 = s]$. We also adopt the conventional definition for Q-functions w.r.t. a policy $\pi$ as $Q^\pi(s, a) = \frac{1}{1-\gamma} \sum_t \mathbb{E}_{a_t \sim \pi(s_t)} [\gamma^t r(s_t, a_t)|s_0 = s, a_0 = a]$, and we use $\hat{Q}^\pi_\theta$ to denote an estimated Q-function (i.e., by a neural network) w.r.t. policy $\pi$.

Given access to an offline dataset collected using a behavior policy $\pi_\beta$, $\mathcal{D} = \{(s, a, r, s')\}$, we aim to first train the best possible policy and value function using the offline dataset $\mathcal{D}$ alone, followed by an online phase that utilizes online interaction in $\mathcal{M}$. Our goal in this fine-tuning phase is to obtain the optimal policy with the fewest number of online samples. This can be expressed as minimizing the **cumulative regret** over rounds of online interaction: $\text{Reg}(T) := \mathbb{E}_{s \sim \rho} \sum_{t=1}^T [V^\star(s) - V^{\pi_t}(s)]$. As we will demonstrate in Section 7, existing methods targeted to this setting often tend to attain regret that shrinks slowly.

Our approach will build upon the conservative Q-learning (CQL) (Kumar et al., 2020) algorithm. CQL imposes an additional regularizer that penalizes the learned Q-function on out-of-distribution (OOD) actions while compensating for this pessimism on actions seen within the training dataset. Assuming that the value function is represented by a function, $Q_\theta$, the training objective of CQL is given by

$$\min_\theta \; \alpha \underbrace{\left(\mathbb{E}_{s \sim \mathcal{D}, a \sim \pi} [Q_\theta(s, a)] - \mathbb{E}_{s, a \sim \mathcal{D}} [Q_\theta(s, a)]\right)}_{\text{Conservative regularizer } \mathcal{R}(\theta)}$$

$$+ \frac{1}{2} \mathbb{E}_{s, a, s' \sim \mathcal{D}} \left[ \left( Q_\theta(s, a) - \mathcal{B}^\pi \bar{Q}(s, a) \right)^2 \right], \quad (3.1)$$

where $\mathcal{B}^\pi \bar{Q}(s, a)$ is the Bellman backup operator applied to a delayed target Q-network, $\bar{Q}$: $\mathcal{B}^\pi \bar{Q}(s, a) := r(s, a) + \gamma \mathbb{E}_{a' \sim \pi(a'|s')}[\bar{Q}(s', a')]$. The second term is the standard TD error (Lillicrap et al., 2015; Fujimoto et al., 2018b; Haarnoja et al., 2018b). The first term $\mathcal{R}(\theta)$ (in blue) is a conservative regularizer that aims to prevent overestimation in the Q-values for OOD actions by minimizing the Q-values under the policy $\pi(a|s)$, which picks actions with high Q-values $Q_\theta(s, a)$, and counterbalances by maximizing the Q-values of the actions in the dataset.

## 4. Which Offline RL Initializations Enable Fast Online Fine-Tuning?

A natural starting point for a method that pre-trains a value function from offline data and then performs online RL fine-tuning is to simply initialize the value function with one that is produced by an existing offline RL method. However, in this section, we will show that initializations learned by many offline RL algorithms tend to perform poorly in online fine-tuning. We will study the reasons for this poor performance for the class of conservative methods to motivate our approach and then use the resulting insights to develop our method, calibrated Q-learning.

### 4.1. Empirical Analysis

To motivate certain conditions that give rise to our method, we seek to understand the limitations of existing offline RL approaches for online fine-tuning. The analysis of Nair et al. (2020b) highlights the limitations of explicit policy constraint methods for fine-tuning, therefore, in this section, we study a representative *implicit* policy constraint method, implicit Q-learning (IQL) (Kostrikov et al., 2021a) that attains good performance on benchmark tasks, and a conservative method, conservative Q-learning (CQL) (Kumar et al., 2020). We study the task of fine-tuning a robot policy on a visual pick-and-place task with a distractor object and sparse binary rewards (the performance of any policy is upper-bounded by +1) from prior work (Singh et al., 2020). More details about the offline dataset are in Appendix C.

We show the learning curves for both methods in online fine-tuning in Figure 2. While the offline Q-function initialization obtained from both approaches attains a similar (normalized) return of around 0.5, neither of these methods performs well during fine-tuning: IQL improves steadily but slowly and cannot outperform CQL asymptotically. Despite the better asymptotic performance, CQL first unlearns the offline initialization and spends about 5K samples to

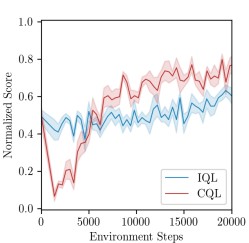

*Figure 2.* **CQL and IQL during online fine-tuning**, which begins at step 0 right after offline training. While CQL suffers from initial policy unlearning, IQL improves slowly throughout fine-tuning.

recover before it begins to improve. This shows how neither of these approaches enables *both* a steady improvement through learning and a better asymptotic performance on this task. In this work, we restrict our focus to developing effective fine-tuning strategies on top of conservative methods. Since these methods already attain good performance asymptotically but exhibit initial unlearning, we wish to now understand the potential reasons behind the initial unlearning in CQL to develop a practical approach. As a side note, we also investigate certain reasons that could explain the speed of IQL in Appendix E.

**Why does CQL unlearn initially?** To understand why this happens, we inspect the Q-values averaged over the dataset

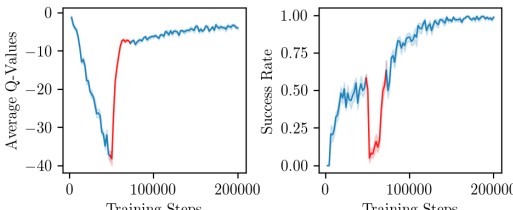

*Figure 3.* **The evolution of the average Q-value and the success rate of CQL over the course of offline pre-training and online fine-tuning.** Fine-tuning begins at 50K steps. The red-colored part denotes the period of performance recovery which also coincides with the period of Q-value adjustment.

in Figure 3. Note that the Q-values learned by CQL in the offline phase are *much* smaller than their ground-truth value as expected, but these Q-values drastically jump and adjust in scale when online fine-tuning begins. In fact, we observe that performance recovery (red segment in Figure 3) coincides with a period where the range of Q-values changes to match the true range. This is expected: as a conservative Q-function experiences new online data actions much worse than the offline initialization on the rollout states appear to attain higher rewards compared to the highly underestimated Q-function initialization, which in turn deceives the policy optimizer into unlearning the initial policy. Once the Q-function has adjusted and the range of Q-values more closely matches the true range, then fine-tuning can proceed normally, after recovering from the dip.

**To summarize,** our analysis indicates that methods such as IQL that are based on policy constraints can lead to slower asymptotic performance. Whereas conservative methods can attain good asymptotic performance, but "waste" samples to correct the learned Q-function. Thus, in this paper we attempt to develop a good fine-tuning method that builds on top of existing conservative offline RL methods (to attain good asymptotic performance), but aims to "calibrate" the Q-function so that the initial dip in performance is avoided.

### 4.2. Conditions That Enable Fast Fine-Tuning

Empirical observations from the preceding discussion motivate two conditions on the offline Q-function initialization to enable fast fine-tuning: **(a)** if the Q-function is **conservative**, then we can attain good asymptotic performance, and **(b)** if the Q-function is **calibrated**, i.e., the learned Q-values are larger than the ground-truth return of policies worse than the offline initialization, then online fine-tuning does not need to devote samples to first unlearn the offline initialization, correct the Q-value scale and recover the performance of the offline initialization again, before fine-tuning proceeds normally. We define this notion of "calibration" below, then present our method to enforce this condition next, and finally, in Section 6 show that this enables fast fine-tuning.

**Definition 4.1** (Calibration). *An estimated Q-function $Q_\theta^\pi$ for a given policy $\pi$ is said to be calibrated with respect to a reference policy $\mu$ if $Q_\theta^\pi(s,a) \geq Q^\mu(s,a), \forall(s,a) \in \mathcal{S} \times \mathcal{A}$.*

If the learned Q-function $Q_\theta$ is calibrated with respect to any policy $\mu$ that performs worse than $\pi$, it would prevent unlearning during fine-tuning that we observed in the case of CQL: the policy optimizer would not unlearn $\pi$ in favor of a worse $\mu$ upon observing new online data since $\pi$ still attains a larger value under the learned $Q_\theta$ function: $Q_\theta(s,a) \geq Q^\mu(s,a)$. However, such a condition is impossible to impose as it requires estimating returns of all policies. Therefore, our approach Cal-QL will enforce calibration only with respect to policies $\mu$ whose Q-value, $Q^\mu(s,a)$, can be estimated reliably (e.g., the behavior policy induced by the dataset). This is the key idea behind our method.

## 5. Cal-QL: Calibrated Q-Learning

Our approach, calibrated Q-learning (Cal-QL), aims to learn conservative and calibrated value function initializations from an offline dataset. To this end, Cal-QL builds on conservative Q-learning (CQL) (Kumar et al., 2020). We then constrain the learned Q-function to produce Q-values larger than the Q-value of a reference policy per Definition 4.1. In principle, our approach can utilize many different choices of reference policies, but for developing a practical method, we simply utilize the behavior policy as our reference policy.

**Estimating $Q^\mu(s,a)$.** We first need to devise a method for obtaining an estimator of $Q^\mu(s,a)$ and a way to constrain $Q_\theta^\pi(s,a)$, the conservative Q-function learned by CQL to $Q^\mu$. For estimating $Q^\mu(s,a)$, we utilize the Monte-Carlo return estimator for environments that end in a terminal and fit a function approximator $Q_\theta^\mu$ to the return-to-go estimators via supervised regression for other environments.

**Incorporating calibration into CQL.** Next, we find a way to constrain the learned Q-function $Q_\theta^\pi$ to be larger than $Q^\mu$. This property can be enforced via a simple change to Equation 3.1 by masking out the push down of the learned Q-value on out-of-distribution (OOD) actions in CQL (i.e., $a$ sampled from the learned policy) if the Q-function is not calibrated, i.e., if $Q_\theta^\pi(s,a) \leq Q^\mu(s,a)$. Cal-QL modifies the CQL regularizer, $\mathcal{R}(\theta)$ in this manner:

$$\mathbb{E}_{s\sim\mathcal{D},a\sim\pi}\left[\max\left(Q_\theta, Q^\mu\right)\right] - \mathbb{E}_{s,a\sim\mathcal{D}}\left[Q_\theta(s,a)\right], \quad (5.1)$$

where the changes from standard CQL are depicted in red. As long as $\alpha$ (in Equation 3.1) is large, for any state-action pair where the learned Q-value is smaller than $Q^\mu$, the Q-function learned by Equation 5.1 is guaranteed to be upper bound $Q^\mu$ in a tabular setting. Of course, as with any practical RL method, with function approximators and gradient-based optimizers, we cannot guarantee that we can enforce this condition for every state-action pair, but in our experiments, we find that Equation 5.1 is sufficient to enforce the calibration in expectation over the states in the dataset.

**Pseudo-code and implementation details.** Our implementation of Cal-QL directly builds on the implementation of

CQL from Geng (2022). We present a pseudo-code for Cal-QL in Algorithm 1. Additionally, we list the hyper-parameters $\alpha$ for the CQL algorithm and our baselines for each suite of tasks in Appendix D. Following the protocol in prior work (Kostrikov et al., 2021a), the practical implementation of Cal-QL trains on a mixture of the offline data and the new online data, weighted in some proportion during fine-tuning. We show in our experiments in Section 7, how this simple *one-line* code change to the training objective drastically improves fine-tuning results compared to prior methods, while being grounded in theory.

# 6. Theoretical Analysis of Cal-QL

In this section, we will analyze the cumulative regret attained over the course of online fine-tuning, when the value function is pre-trained with Cal-QL, and show that enforcing both conservatism and calibration (Defintion 4.1) leads to a favorable regret bound during the online phase compared to utilizing naïve uncalibrated conservative methods. Our analysis utilizes tools from Song et al. (2022), but analyzes a different algorithm that runs conservative offline RL with calibration for offline training and online fine-tuning.

**Notation & terminology.** To set up notation and terminology for our analysis, we will consider an idealized version of Cal-QL for simplicity. Specifically, following prior work (Song et al., 2022), we will operate in a finite-horizon setting with a horizon $H$. We denote the learned Q-function at any learning iteration $t$, for a given $(s, a)$ pair and time-step $h$ by $Q_\theta^t(s, a)$. For any given policy $\pi$, let $C_\pi \geq 1$ denote the concentrability coefficient, i.e., a coefficient that quantifies the distribution shift between the policy $\pi$ and the dataset $\mathcal{D}$, in terms of the ratio of Bellman errors averaged under $\pi$ and the dataset $\mathcal{D}$. We also use $C_\pi^\mu$ to denote the concentrability coefficient over a subset of the Q-function class induced by a reference policy $\mu$, which intuitively provides $C_\pi^\mu \leq C_\pi$. Let $d_\mu$ denote the intrinsic dimension of $\mathcal{C}_\mu$, the Q-function class w.r.t. the reference policy $\mu$ and let $d$ denote the intrinsic dimension of $\mathcal{C}$, the Q-function class w.r.t. all policies $\pi$. Intuitively, we have $\mathcal{C}_\mu \subset \mathcal{C}$, which implies that $d_\mu < d$. The formal definitions are provided in Appendix B.2. This version of Cal-QL will first run fitted Q-iteration while enforcing an upper bound on the difference between Q-values at policy actions and dataset actions as a constraint, and then impose Condition 4.1 against the Q-function of a reference policy, $Q^\mu(s, a)$, which we assume is known apriori for simplicity of our analysis. We will use $\pi^t$ to denote the arg-max policy induced by $Q^t$.

We first informally discuss our proof insight, aiming to intuitively convey how calibration and conservatism enable Cal-QL to attain a smaller regret compared to not imposing either condition. Then, we present our main formal bound.

**Intuition.** Our goal is to bound the cumulative regret of online fine-tuning, $\mathrm{Reg}(T) = \sum_t \mathbb{E}_{s_0 \sim \rho}[V^{\pi^\star}(s_0) - V^{\pi^t}(s_0)]$.

We can decompose this expression of regret into two terms:

$$\mathrm{Reg}(T) = \underbrace{\sum_{t=1}^{T} \mathbb{E}_{s_0 \sim \rho} \left[ V^\star(s_0) - \max_a Q_\theta^t(s_0, a) \right]}_{(i) := \text{ miscalibration}}$$
$$+ \underbrace{\sum_{t=1}^{T} \mathbb{E}_{s_0 \sim \rho} \left[ \max_a Q_\theta^t(s_0, a) - V^{\pi^t}(s_0) \right]}_{(ii) := \text{ optimism}}. \quad (6.1)$$

This decomposition of regret into terms (i) and (ii) is instructive. Term (ii) corresponds to the amount of over-estimation in the learned value function, which is expected to be small if a conservative RL algorithm is used for training. Term (i) is the difference between the ground-truth value of the optimal policy and the learned Q-function and is negative if the learned Q-function were calibrated against the optimal policy (per Definition 4.1). Of course, this is not always possible, but note that when Cal-QL utilizes a reference policy $\mu$ with a high value $V^\mu$, close to $V^\star$, then the learned Q-function $Q_\theta$ is calibrated with respect to $Q^\mu$ per Condition 4.1 and term (i) can still be controlled. Therefore, controlling this regret requires striking a balance between learning a calibrated (term (i)) and conservative (term (ii)) Q-function. We now formalize this intuition.

**Theorem 6.1** (Informal regret bound of Cal-QL). *With high probability, Cal-QL obtains the following bound on total regret accumulated during online fine-tuning:*
$$\widetilde{O} \left( \min \left\{ C_{\pi^\star}^\mu H \sqrt{dT}, \ T \mathbb{E}_\rho[V^\star(s_0) - V^\mu(s_0)] + H \sqrt{d_\mu T} \right\} \right).$$
A formal version and the proof is provided in Appendix B.4.

**Comparison to Song et al. (2022).** Song et al. (2022) analyzes an online RL algorithm that utilizes offline data without imposing conservatism or calibration. We now compare Theorem 6.1 to Theorem 1 of Song et al. (2022) to understand the impact of these conditions on the final regret guarantee. Theorem 1 of Song et al. (2022) presents a regret bound: $\mathrm{Reg}(T) = \widetilde{O} \left( C_{\pi^\star} H \sqrt{dT} \right)$ and we note some improvements in our guarantee, that we also verify via experiments in Section 7.2: **(a)** for the setting when the reference policy $\mu$ is the behavior policy, and this policy exhibits near-optimal behavior, i.e., $V^\star - V^\mu \lesssim O(H\sqrt{d/T})$, then Cal-QL can enable a tighter regret guarantee compared to Song et al. (2022); **(b)** as we show in Appendix B.3, the concentrability coefficient $C_{\pi^\star}^\mu$ appearing in our guarantee is already smaller than the one that appears in Theorem 1 of Song et al. (2022), providing another source of improvement; and **(c)** finally, in the worst possible case, where the reference policy is diverse and highly sub-optimal, Cal-QL reverts back to the guarantee from Song et al. (2022) meaning that Cal-QL is not any worse than this prior work.

# 7. Experimental Evaluation

The goal of our experimental evaluation is to study how well Cal-QL can learn value functions from offline data

that facilitate sample-efficient online fine-tuning. To this end, we study the performance of Cal-QL in comparison with several other state-of-the-art methods on a variety of offline RL benchmark tasks from D4RL (Fu et al., 2020) and Nair et al. (2020b), evaluating performance before and after online fine-tuning. We also study the effectiveness of Cal-QL on higher-dimensional tasks, where the policy and value function must process raw image observations. Finally, we perform several empirical studies to understand the efficacy of Cal-QL with different dataset compositions and to understand the impact of errors in reference function value estimation on Cal-QL.

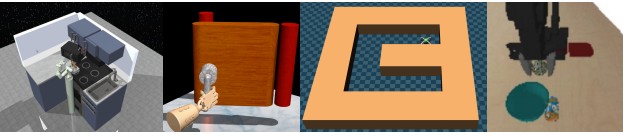

Figure 4. **Tasks:** We evaluate Cal-QL on a diverse set of benchmark problems: `Frankakitchen` and `AntMaze` domains from Fu et al. (2020), `Adroit` tasks from Nair et al. (2020b) and a vision-based robotic manipulation task from Kumar et al. (2022).

**Offline RL tasks and datasets.** We evaluate Cal-QL on a number of benchmark tasks and datasets, previously used by prior works (Kostrikov et al., 2021a; Nair et al., 2020b) to evaluate fine-tuning performance: **(1)** the `AntMaze` tasks from D4RL (Fu et al., 2020) that require controlling an 8-DoF ant quadruped robot to navigate from a starting point to a desired goal location in a maze. The reward is +1 if the agent reaches within a pre-specified small radius around the goal and 0 otherwise. We consider two kinds of maze layouts (medium and large mazes from Fu et al. (2020)) and two data compositions: **play** and **diverse** that vary in coverage of actions at different regions of the state space and sub-optimality of the behavior policy; **(2)** the `FrankaKitchen` tasks from D4RL require controlling a 9-DoF Franka robot to attain a desired configuration of a kitchen. To succeed, a policy must complete four subtasks in the kitchen within a single rollout, and it receives a binary reward of +1 for every sub-task it completes; **(3)** three `Adroit` dexterous manipulation tasks (Rajeswaran et al., 2018; Kostrikov et al., 2021a; Nair et al., 2020b) that require learning complex manipulation skills on a 28-DoF five-fingered hand to **(a)** manipulate a pen in-hand to a desired configuration (**pen-binary**), **(b)** open a door by unlatching the handle (**door-binary**), and **(c)** relocating a ball to a desired location (**relocate-binary**). An agent obtains a sparse binary +1/0 reward if it succeeds in solving the task. Each of these tasks only provides an extremely narrow offline dataset consisting of 25 demonstrations collected via human teleoperation. Finally to evaluate the efficacy of Cal-QL on more challenging tasks where we must learn from raw visual observations, we study **(4)** a pick-and-place task from prior work (Kumar et al., 2022; Singh et al., 2020) that requires learning to pick a ball and place it in a bowl, in the presence of distractors.

**Comparisons, prior methods, and evaluation protocol.** We compare Cal-QL to running online SAC (Haarnoja et al., 2018c) from scratch, as well as prior approaches that leverage offline data. This includes fine-tuning with CQL (Kumar et al., 2020) after offline pre-training, as well as with IQL (Kostrikov et al., 2021a), which were specifically proposed as effective offline pre-training / online fine-tuning methods. We also compare to a baseline that trains SAC (Haarnoja et al., 2018c) using both online data and offline data (denoted by "SAC + offline data"). This simple and pragmatic approach most closely matches DDPGfD (Vecerik et al., 2017), updated to use the stronger online off-policy RL algorithm (SAC). Note that in contrast to DDPGfD, we are using the offline data provided in each benchmark task, which is *not necessarily demonstration* data. We present learning curves for online fine-tuning and also quantitatively evaluate each method on its ability to improve the initialization learned from offline data measured in terms of final performance after a pre-defined number of steps per domain, as well as the cumulative regret accumulated over the course of online fine-tuning which measures the speed of fine-tuning over the course of online interaction.

### 7.1. Empirical Results

We first present a quantitative comparison of Cal-QL in terms of the normalized performance obtained before and after fine-tuning in Table 1 and the cumulative regret accumulated in a fixed number of steps of environment interaction in Table 2. Following the protocol of Fu et al. (2020), we normalize the average return values for each domain with respect to the highest possible return (+4 in FrankaKitchen; +1 in other tasks; see Appendix D.1 for more details).

**Cal-QL improves the offline initialization significantly.** Observe in Table 1 and Figure 5 that while the performance of offline initialization acquired by Cal-QL performs about comparably (or slightly worse) to the initialization acquired by other methods such as IQL, Cal-QL is able to improve over its offline initialization by over **2x** outperforming the next best method (IQL and CQL) on **8/11** tasks, by about **32%** in terms of aggregate performance after fine-tuning.

**Cal-QL enables fast fine-tuning.** To understand the efficacy of Cal-QL in enabling learning quickly during online fine-tuning, we measure the cumulative regret accumulated over the course of fine-tuning. Observe in Table 2 that Cal-QL consistently achieves a smaller regret of 0.22 on **9/11** tasks, improving over the next best method by **43%**. In tasks such as `relocate-binary`, Cal-QL enjoys the fast online learning benefits associated with naïve online RL methods that incorporate the offline data in the replay buffer (SAC + offline data and Cal-QL are the only two methods to attain a score of $\geq 99\%$ on this task) unlike prior

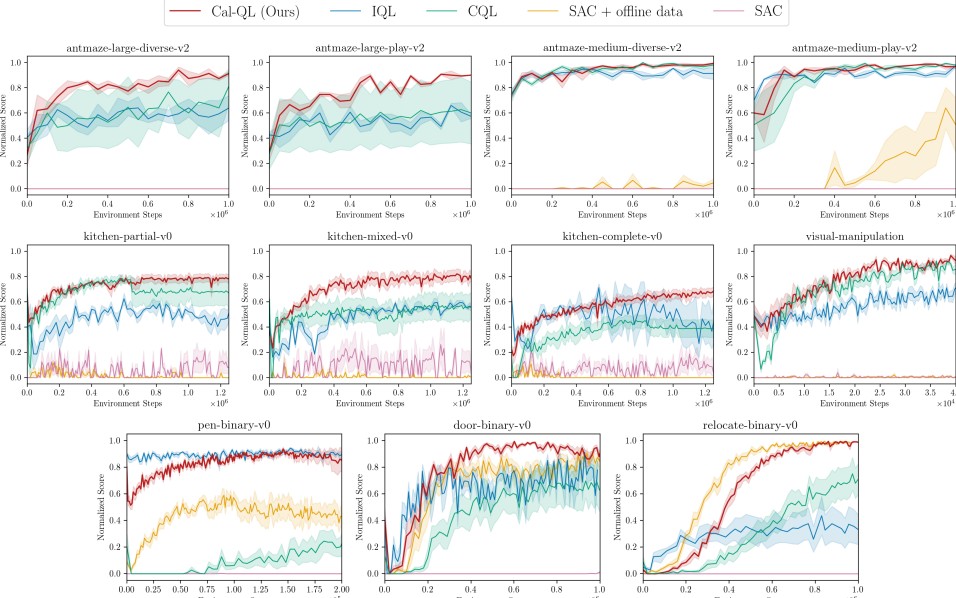

*Figure 5.* **Online fine-tuning after offline initialization on the benchmark tasks**. The plots show the online fine-tuning phase *after* pre-training for each method (except SAC-based approaches which are not pre-trained). We use more than 3 seeds for each method. Observe that Cal-QL consistently matches or exceeds the learning speed and final performance of the best prior method and is the only algorithm to do so across all tasks.

offline RL methods such as IQL. As shown in Figure 5, in the `kitchen` and `antmaze` domains, Cal-QL brings the benefits of fast online learning together with a good offline initialization, improving drastically on the regret metric. Finally, observe that the initial unlearning at the beginning of fine-tuning with conservative methods observed in Section 4.1 is greatly alleviated in all tasks: especially note that this unlearning disappears completely on the visual pick and place task studied previously in Section 4.1.

### 7.2. Understanding the Behavior of Cal-QL

In this section, we aim to understand the behavior of Cal-QL by performing controlled experiments that modify the dataset composition, and by investigating various metrics to understand the properties of scenarios where utilizing Cal-QL is especially important.

**Effect of data composition.** To understand the efficacy of Cal-QL with different data compositions, we ran it on a newly constructed fine-tuning task on the medium-size `AntMaze` domain with a low-coverage offline dataset, which is generated via a scripted controller that starts from a fixed initial position and navigates the ant to a fixed goal position. In Figure 6, we plot the performance of Cal-QL and baseline CQL (for comparison) on this task, alongside the trend of average Q-values over the course of offline pre-training (to the left of the dashed vertical line, before 250 training epochs) and online fine-tuning (to the right of the vertical dashed line, after 250 training epochs), and the trend of *bounding rate*, i.e., the fraction of transitions in the data-buffer for which the constraint in Cal-QL actively

lower-bounds the learned Q-function with the reference Q-value. For comparison, we also plot these quantities for a diverse dataset with high coverage on the task (we use the antmaze-medium-diverse-v2 from Fu et al. (2020) as a representative diverse dataset) in Figure 6.

Observe that for the diverse dataset, both naïve CQL and Cal-QL perform similarly, and indeed, the learned Q-values behave similarly for both of these methods. In this setting, online learning doesn't waste any samples to correct the Q-function when fine-tuning begins leading to a low bounding rate, almost always close to 0. Instead, with the narrow dataset, we observe that the Q-values learned by naïve CQL are much smaller, and are corrected once fine-tuning begins. This correction co-occurs with a drop in performance (solid blue line on left), and naïve CQL is unable to recover from this drop. Cal-QL which calibrates the scale of the Q-function by lower bounding the Q-values for many more samples in the dataset, stably transitions to online fine-tuning phase with no unlearning (solid red line on left).

The study reveals the behavior of Cal-QL: in settings with narrow datasets (e.g., in the experiment above and in the `adroit` and `visual-manipulation` domains from Figure 5), Q-values learned by naïve conservative methods are more likely to be smaller than the ground-truth Q-function of the behavior policy due to function approximation errors, in which case utilizing Cal-QL to calibrate the Q-function against the behavior policy can be significantly helpful. On the other hand, with significantly high-coverage datasets, especially in problems where the behavior policy

| Domain | Task | IQL | CQL | SAC + offline data | SAC | Cal-QL (Ours) |
|---|---|---|---|---|---|---|
| antmaze | large-diverse | $0.41 \rightarrow 0.64$ | $0.32 \rightarrow 0.81$ | $0.00 \rightarrow 0.00$ | $0.00 \rightarrow 0.00$ | $0.27 \rightarrow \mathbf{0.91}$ |
| | large-play | $0.43 \rightarrow 0.57$ | $0.28 \rightarrow 0.60$ | $0.00 \rightarrow 0.00$ | $0.00 \rightarrow 0.00$ | $0.29 \rightarrow \mathbf{0.90}$ |
| | medium-diverse | $0.75 \rightarrow 0.91$ | $0.74 \rightarrow 0.98$ | $0.00 \rightarrow 0.05$ | $0.00 \rightarrow 0.00$ | $0.74 \rightarrow \mathbf{0.99}$ |
| | medium-play | $0.70 \rightarrow 0.97$ | $0.51 \rightarrow \mathbf{0.98}$ | $0.00 \rightarrow 0.51$ | $0.00 \rightarrow 0.00$ | $0.60 \rightarrow 0.97$ |
| kitchen | partial | $0.40 \rightarrow 0.47$ | $0.70 \rightarrow 0.68$ | $0.00 \rightarrow 0.00$ | $0.00 \rightarrow 0.08$ | $0.66 \rightarrow \mathbf{0.78}$ |
| | mixed | $0.48 \rightarrow 0.57$ | $0.62 \rightarrow 0.56$ | $0.00 \rightarrow 0.01$ | $0.00 \rightarrow 0.12$ | $0.37 \rightarrow \mathbf{0.81}$ |
| | complete | $0.63 \rightarrow 0.37$ | $0.14 \rightarrow 0.39$ | $0.00 \rightarrow 0.00$ | $0.00 \rightarrow 0.09$ | $0.22 \rightarrow \mathbf{0.67}$ |
| adroit | pen-binary | $0.90 \rightarrow \mathbf{0.90}$ | $0.20 \rightarrow 0.20$ | $0.09 \rightarrow 0.43$ | $0.00 \rightarrow 0.00$ | $0.80 \rightarrow 0.85$ |
| | door-binary | $0.42 \rightarrow 0.88$ | $0.13 \rightarrow 0.63$ | $0.00 \rightarrow 0.93$ | $0.00 \rightarrow 0.00$ | $0.40 \rightarrow \mathbf{0.94}$ |
| | relocate-binary | $0.04 \rightarrow 0.34$ | $0.09 \rightarrow 0.67$ | $0.00 \rightarrow \mathbf{1.00}$ | $0.00 \rightarrow 0.00$ | $0.03 \rightarrow 0.99$ |
| COG | visual-manipulation | $0.49 \rightarrow 0.70$ | $0.50 \rightarrow 0.86$ | $0.00 \rightarrow 0.01$ | $0.00 \rightarrow 0.00$ | $0.49 \rightarrow \mathbf{0.93}$ |
| | average | $0.51 \rightarrow 0.67\ (+31.3\%)$ | $0.38 \rightarrow 0.67\ (+76.3\%)$ | $N/A \rightarrow 0.27$ | $N/A \rightarrow 0.03$ | $0.44 \rightarrow \mathbf{0.89}\ (+102.3\%)$ |

*Table 1.* **Normalized score before and after the online fine-tuning.** We trained each method for 1M environment steps on `antmaze`, `door-binary`, and `relocate-binary` tasks, 200K steps on `pen-binary`, 1.25M steps on `kitchen` tasks, and 40K steps on `visual-manipulation`. Observe that Cal-QL improves over the best prior fine-tuning method and attains a much larger performance improvement over the course of online fine-tuning.

| Task | IQL | CQL | SAC+od | SAC | Cal-QL |
|---|---|---|---|---|---|
| large-diverse | 0.43 | 0.41 | 1.00 | 1.00 | **0.21** |
| large-play | 0.47 | 0.45 | 1.00 | 1.00 | **0.24** |
| medium-diverse | 0.09 | **0.06** | 0.99 | 1.00 | **0.06** |
| medium-play | 0.10 | 0.12 | 0.84 | 1.00 | **0.09** |
| partial | 0.53 | 0.33 | 0.98 | 0.94 | **0.27** |
| mixed | 0.53 | 0.49 | 0.99 | 0.90 | **0.29** |
| complete | 0.52 | 0.65 | 0.99 | 0.93 | **0.44** |
| pen-binary | **0.10** | 0.91 | 0.56 | 1.00 | 0.16 |
| door-binary | 0.37 | 0.53 | 0.93 | 1.00 | **0.18** |
| relocate-binary | 0.70 | 0.67 | **0.20** | 1.00 | 0.28 |
| visual-manipulation | 0.43 | 0.30 | 1.00 | 1.00 | **0.23** |
| average | 0.39 | 0.45 | 0.86 | 0.98 | **0.22** |

*Table 2.* **Cumulative regret averaged over the steps of online fine-tuning.** The smaller the better, worst case is 1.00. Note that Cal-QL attains the smallest regret in aggregate, improving over the best prior method in 9/11 tasks that we study.

is also random and sub-optimal, Q-values learned by naïve methods are likely to already be calibrated with respect to those of the behavior policy, and no explicit calibration might be needed (and indeed, the bounding rate tends to be very close to 0 in Figure 6). In this case, Cal-QL will revert back to standard CQL, as we observe in the case of the diverse dataset above. This intuition is also reflected in Theorem 6.1: when the reference policy $\mu$ is close to a narrow, expert policy, we would expect Cal-QL to be especially effective in controlling the efficiency of online fine-tuning.

**Estimation errors in the reference value function do not affect performance significantly.** In our experiments, we compute the reference value functions using Monte-Carlo return estimates. However, this may not be available in all tasks. How does Cal-QL behave when reference value functions must be estimated using the offline dataset itself?

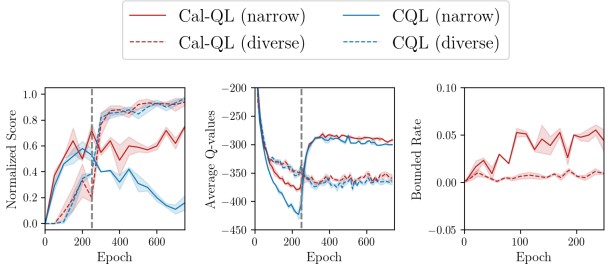

*Figure 6.* **Performance of Cal-QL with different data compositions.** Cal-QL is most effective with narrow datasets, where Q-values need to be corrected at the beginning of fine-tuning.

To answer this, we ran an experiment on the `kitchen` domain, where instead of using an estimate for $Q^\mu$ based on the Monte-Carlo return, we train a neural network function approximator $Q^\mu_\theta$ to approximate $Q^\mu$ via supervised regression on to Monte-Carlo return, which is then utilized by Cal-QL. Observe in Figure 7, that the performance of Cal-QL largely remains unaltered. This implies as long as we can obtain a reasonable function approximator to estimate the Q-function of the reference policy (in this case, the behavior policy), errors in this reference Q-function do not affect the performance of Cal-QL significantly.

## 8. Discussion

In this work we developed, Cal-QL a method for acquiring offline initializations that facilitate fast online fine-tuning. Cal-QL learns conservative value functions, but additionally constrains to be larger than the value function of a reference policy. This form of calibration of the Q-function allows us to avoid initial unlearning in online fine-tuning with conservative methods, while also retaining effective asymptotic performance that these methods typically exhibit. Our theoretical and experimental results

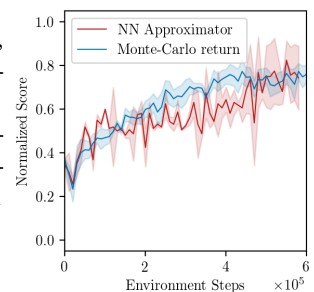

*Figure 7.* **The performance of Cal-QL using a neural network approximator for the reference value function is comparable to using the Monte-Carlo return.** This indicates that errors in the reference Q-function do not negatively impact the online fine-tuning performance.

highlight the benefit of Cal-QL in enabling fast online fine-tuning. While Cal-QL already outperforms prior methods, we believe there is a scope for developing even more practically effective methods by carefully adjusting calibration and conservatism (see Theorem 6.1). Another interesting direction for future work is to extend Cal-QL to settings with different pre-training and fine-tuning tasks that appear real-world problems.

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

# Appendices

## A. Implementation details of Cal-QL

Our algorithm, Cal-QL is illustrated in Algorithm 1. A complete implementation of the functions in python-style is provided in Appendix A.2.

### A.1. Cal-QL Algorithm

---
**Algorithm 1** Cal-QL pseudo-code

---
1: Initialize Q-function, $Q_\theta$, a policy, $\pi_\phi$
2: **for** step $t$ in $\{1, \ldots, N\}$ **do**
3:     Train the Q-function using $J_Q(\theta)$ in Eq. 5.1:

$$\theta_t := \theta_{t-1} - \eta_Q \nabla_\theta J_Q(\theta) \qquad (A.1)$$

4:     Improve policy $\pi_\phi$ with SAC-style update:

$$\phi_t := \phi_{t-1} + \eta_\pi \mathbb{E}_{s \sim \mathcal{D}, a \sim \pi_\phi(\cdot|s)}[Q_\theta(s, a) - \log \pi_\phi(a|s)] \qquad (A.2)$$

5:     If the online phase begins, change CQL regularizer coefficient $\alpha$
    from $\alpha_{\text{offline}}$ to $\alpha_{online}$.
6: **end for**

---

### A.2. Python Implementation

*Listing 1.* Training Q networks given a batch of data

```python
cql_alpha = self.online_alpha if training_phase == 'online' else self.offline_alpha
q_data = critic(batch['observations'], batch['actions'])

next_dist = actor(batch['next_observations'])
next_pi_actions, next_log_pis = next_dist.sample()

target_qval = target_critic(batch['observations'], next_pi_actions)
target_qval = batch['rewards'] + self.gamma * (1 - batch['dones']) * target_qval

td_loss = mse_loss(q_data, target_qval)

num_samples = 4
random_actions = uniform((num_samples, batch_size, action_dim), min=-1, max=1)
random_pi = 0.5 ** batch['actions'].shape[-1]

pi_actions, log_pis = actor(batch['observations'])

q_rand_is = critic(batch['observations'], random_actions) - random_pi
q_pi_is = critic(batch['observations'], pi_actions) - log_pis

mc_return = batch['mc_return'].repeat(num_samples)
q_rand_is = max(q_rand_is, mc_return)
q_pi_is = max(q_rand_is, mc_return)

cat_q = concatenate([q_rand_is, q_pi_is], new_axis=True)
cat_q = logsumexp(cat_q, axis=0) # sum over num_samples
critic_loss = td_loss + ((cat_q - q_data).mean() * cql_alpha)

critic_optimizer.zero_grad()
critic_loss.backward()
critic_optimizer.step()
```

*Listing 2.* Training the policy (or the actor) given a batch of data

```
# return distribution of actions
pi_actions, log_pis = actor(batch['observations'])

# calculate q value of actor actions
qpi = critic(batch['observations'], actions)
qpi = qpi.min(axis=0)

# same objective as CQL (kumar et al.)
actor_loss = (log_pis * self.alpha - qpi).mean()

# optimize loss
actor_optimizer.zero_grad()
actor_loss.backward()
actor_optimizer.step()
```

# B. Regret Analysis of Cal-QL

We provide a theoretical version of Cal-QLin Algorithm 2. Policy fine-tuning has been studied in different settings (Xie et al., 2021; Song et al., 2022; Wagenmaker & Pacchiano, 2022). Our analysis largely adopts the settings and results in Song et al. (2022), with additional changes in Assumption B.1, Assumption B.2, and Definition B.3. Note that the goal of this proof is to demonstrate that *a pessimistic functional class (Assumption B.1)* allows one to utilize the offline data efficiently, rather than providing a new analytical technique for regret analysis. See comparisons between Section B.3 and Section G.1. Note that we use $f$ instead of $Q_\theta$ in the main text to denote the estimated $Q$ function for notation simplicity.

---

**Algorithm 2** Theoretical version of Cal-QL

1: **Input:** Value function class $\mathcal{F}$, # total iterations $T$, offline dataset $\mathcal{D}_h^\nu$ of size $m_{\text{off}}$ for $h \in [H-1]$.
2: Initialize $f_h^1(s,a) = 0, \forall(s,a)$.
3: **for** $t = 1, \ldots, T$ **do**
4:      Let $\pi^t$ be the greedy policy w.r.t. $f^t$                 $\triangleright$ I.e., $\pi_h^t(s) = \arg\max_a f_h^t(s,a)$.
5:      For each $h$, collect $m_{\text{on}}$ online tuples $\mathcal{D}_h^t \sim d_h^{\pi^t}$           $\triangleright$ online data collection
6:      Set $f_H^{t+1}(s,a) = 0, \forall(s,a)$.
7:      **for** $h = H-1, \ldots 0$ **do**                         $\triangleright$ FQI with offline and online data
8:          Estimate $f_h^{t+1}$ using conservative least squares on the aggregated data:     $\triangleright$ I.e., CQL regularized class $\mathcal{C}_h$

$$f_h^{t+1} \leftarrow \arg\min_{f \in \mathcal{C}_h} \left\{ \widehat{\mathbb{E}}_{\mathcal{D}_h^\nu} \left[ f(s,a) - r - \max_{a'} f_{h+1}^{t+1}(s',a') \right]^2 + \sum_{\tau=1}^t \widehat{\mathbb{E}}_{\mathcal{D}_h^\tau} \left[ f(s,a) - r - \max_{a'} f_{h+1}^{t+1}(s',a') \right]^2 \right\} \quad \text{(B.1)}$$

9:          $f_h^{t+1} = \max\{f_h^{t+1}, Q_h^{\text{ref}}\}$                    $\triangleright$ Set the return of a reference policy as lower bound
10:      **end for**
11: **end for**
12: **Output:** $\pi^T$

---

## B.1. Preliminaries

In this subsection, we follow most of the notations and definitions in Song et al. (2022). In particular, we consider the finite horizon cases, where the value function and Q function are defined as:

$$V_h^\pi(s) = \mathbb{E}\left[ \sum_{\tau=h}^{H-1} r_\tau | \pi, s_h = s \right] \quad \text{(B.2)}$$

$$Q_h^\pi(s,a) = \mathbb{E}\left[ \sum_{\tau=h}^{H-1} r_\tau | \pi, s_h = s, a_h = a \right]. \quad \text{(B.3)}$$

We also define the Bellman operator $\mathcal{T}$ such that $\forall f : \mathcal{S} \times \mathcal{A}$:

$$\mathcal{T}f(s,a) = \mathbb{E}_{s,a}[R(s,a)] + \mathbb{E}_{s' \sim P(s,a)} \max_{a'} f(s', a'), \ \forall (s,a) \in \mathcal{S} \times \mathcal{A}, \tag{B.4}$$

where $R(s,a) \in \Delta[0,1]$ represents a stochastic reward function.

### B.2. Notations

- Feature covariance matrix $\Sigma_{t;h}$:

$$\mathbf{\Sigma}_{t;h} = \sum_{\tau=1}^{t} X_h(f^\tau)(X_h(f^\tau))^\top + \lambda \mathbf{I} \tag{B.5}$$

- Matrix Norm Zanette et al. (2021): for a matrix $\Sigma$, the matrix norm $\|\mathbf{u}\|_{\mathbf{\Sigma}}$ is defined as:

$$\|\mathbf{u}\|_{\mathbf{\Sigma}} = \sqrt{\mathbf{u}\mathbf{\Sigma}\mathbf{u}^\top} \tag{B.6}$$

- Weighted $\ell^2$ norm: for a given distribution $\beta \in \Delta(\mathcal{S} \times \mathcal{A})$ and a function $f : \mathcal{S} \times \mathcal{A} \mapsto \mathbb{R}$, we denote the weighted $\ell^2$ norm as:

$$\|f\|_{2,\beta}^2 := \sqrt{\mathbb{E}_{s,a \sim \beta} f^2(s,a)} \tag{B.7}$$

- A stochastic reward function $R(s,a) \in \Delta([0,1])$

- For each offline data distribution $\nu = \{\nu_0, \ldots, \nu_{H-1}\}$, the offline data set at time step $h$ ($\nu_h$) contains data samples $(s,a,r,s')$, where $(s,a) \sim \nu_h, r \in R(s,a), s' \sim P(s,a)$.

- Given a policy $\pi := \{\pi_0, \ldots, \pi_{H-1}\}$, where $\pi_h : \mathcal{S} \mapsto \Delta(\mathcal{A})$, $d_h^\pi \in \Delta(s,a)$ denotes the state-action occupancy induced by $\pi$ at step $h$.

- We consider the value-based function approximation setting, where we are given a function class $\mathcal{C} = \mathcal{C}_0 \times \ldots \mathcal{C}_{H-1}$ with $\mathcal{C}_h \subset \mathcal{S} \times \mathcal{A} \mapsto [0, V_{\max}]$.

- A policy $\pi^f$ is defined as the greedy policy w.r.t. $f$: $\pi_h^f(s) = \arg\max_a f_h(s,a)$. Specifically, at iteration $t$, we use $\pi^t$ to denote the greedy policy w.r.t. $f^t$.

### B.3. Assumptions and Defintions

**Assumption B.1** (Pessimistic Realizability and Completeness). *For any policy $\pi^e$, we say $\mathcal{C}_h$ is a pessimistic function class w.r.t. $\pi^e$, if for any $h$, we have $Q_h^{\pi^e} \in \mathcal{C}_h$, and additionally, for any $f_{h+1} \in \mathcal{C}_{h+1}$, we have $\mathcal{T}f_{h+1} \in \mathcal{C}_h$ and $f_h(s,a) \leq Q_h^{\pi^e}(s,a), \forall (s,a) \in \mathcal{S} \times \mathcal{A}$.*

**Assumption B.2** (Bilinear Rank of Reference Policies). *Suppose $Q^{\mathrm{ref}} \in \mathcal{C}_{\mathrm{ref}} \subset \mathcal{C}$, where $\mathcal{C}_{\mathrm{ref}}$ is the function class of our reference policy, we assume the Bilinear rank of $\mathcal{C}_{\mathrm{ref}}$ is $d_{\mathrm{ref}}$ and $d_{\mathrm{ref}} \leq d$.*

**Definition B.3** (Bellman error transfer coefficient). *For any policy $\pi$, we define the transfer coefficient on $\mathcal{C}$ as*

$$C_\pi := \max_{f \in \mathcal{C}} \frac{\sum_{h=0}^{H-1} \mathbb{E}_{s,a \sim d_h^\pi}[\mathcal{T}f_{h+1}(s,a) - f_h(s,a)]}{\sqrt{\sum_{h=0}^{H-1} \mathbb{E}_{s,a \sim \nu_h}(\mathcal{T}f_{h+1}(s,a) - f_h(s,a))^2}}. \tag{B.8}$$

**Definition B.4** (Calibrated Bellman error transfer coefficient). *For any policy $\pi$, we define the calibrated transfer coefficient w.r.t. to a reference policy $\pi^{\mathrm{ref}}$ as*

$$C_\pi^{\mathrm{ref}} := \max_{f \in \mathcal{C}, f(s,a) \geq Q^{\mathrm{ref}}(s,a)} \frac{\sum_{h=0}^{H-1} \mathbb{E}_{s,a \sim d_h^\pi}[\mathcal{T}f_{h+1}(s,a) - f_h(s,a)]}{\sqrt{\sum_{h=0}^{H-1} \mathbb{E}_{s,a \sim \nu_h}(\mathcal{T}f_{h+1}(s,a) - f_h(s,a))^2}}, \tag{B.9}$$

*where $Q^{\mathrm{ref}} = Q^{\pi^{\mathrm{ref}}}$.*

By the definition of $C_\pi^{\mathrm{ref}}$ and $C_\pi$, we naturally have $C_\pi^{\mathrm{ref}} \leq C_\pi$.

## B.4. Our Results

**Theorem B.5** (Formal Result of Theorem 6.1). *Fix $\delta \in (0, 1), m_{\mathrm{off}} = T, m_{\mathrm{on}} = 1$, suppose and the function class $\mathcal{C}$ follows Assumption B.1 w.r.t. $\pi^e$. Suppose the underlying MDP admits Bilinear rank $d$ on function class $\mathcal{C}$ and $d_{\mathrm{ref}}$ on $\mathcal{C}_{\mathrm{ref}}$, respectively, then with probability at least $1 - \delta$, Algorithm 2 obtains the following bound on cumulative suboptimality w.r.t. any comparator policy $\pi^e$:*

$$\sum_{t=1}^{T} V^{\pi^e} - V^{\pi^t} = \widetilde{O}\left(\min\left\{C_{\pi^e}^{\mathrm{ref}} H\sqrt{dT},\ T\left(V^{\pi^e} - V^{\mathrm{ref}}\right) + H\sqrt{d_{\mathrm{ref}}T}\right\}\right) \tag{B.10}$$

*where $T_1 = \sum_{t=1}^{T} \mathbb{1}\left\{f_0^t(s,a) > Q^{\mathrm{ref}}(s,a)\right\}$ and $T_2 = \sum_{t=1}^{T} \mathbb{1}\left\{f_0^t(s,a) \leq Q^{\mathrm{ref}}(s,a)\right\}$.*

Note that Theorem B.5 provides a guarantee for *any* comparator policy $\pi^e$, which can be directly applied to $\pi^\star$ described in our informal result (Theorem 6.1). We also change the notation for the reference policy from $\mu$ in Theorem 6.1 to $\pi^{\mathrm{ref}}$ (similarly, $d_{\mathrm{ref}}, V^{\mathrm{ref}}, C_{\pi^e}^{\mathrm{ref}}$ correspond to $d_\mu, V^\mu, C_{\pi^e}^\mu$ in Theorem 6.1) for notation consistency in the proof. Our proof of Theorem B.5 largely follows the proof of Theorem 1 of (Song et al., 2022), and the major changes are caused by Assumption B.1, Assumption B.2, Definition B.3, and Definition B.4.

*Proof.* Let $V^{\mathrm{ref}}(s) = \max_a Q^{\mathrm{ref}}(s,a)$, we start by noting that

$$\sum_{t=1}^{T} V^{\pi^e} - V^{\pi^{f^t}} = \sum_{t=1}^{T} \mathbb{E}_{s\sim\rho}\left[V_0^{\pi^e}(s) - V_0^{\pi^{f^t}}(s)\right]$$

$$= \underbrace{\sum_{t=1}^{T} \mathbb{E}_{s\sim\rho}\left[\mathbb{1}\left\{\bar{\mathcal{E}}_t\right\}\left(V_0^{\pi^e}(s) - V^{\mathrm{ref}}(s)\right)\right]}_{\Gamma_0} + \underbrace{\sum_{t=1}^{T} \mathbb{E}_{s\sim\rho}\left[\mathbb{1}\left\{\bar{\mathcal{E}}_t\right\}\left(V^{\mathrm{ref}}(s) - \max_a f_0^t(s,a)\right)\right]}_{=0,\text{by the definition of }\bar{\mathcal{E}}_t}$$

$$+ \underbrace{\sum_{t=1}^{T} \mathbb{E}_{s\sim\rho}\left[\mathbb{1}\left\{\bar{\mathcal{E}}_t\right\}\left(\max_a f_0^t(s,a) - V_0^{\pi^{f^t}}(s)\right)\right]}_{\Gamma_1} + \underbrace{\sum_{t=1}^{T} \mathbb{E}_{s\sim\rho}\left[\mathbb{1}\left\{\mathcal{E}_t\right\}\left(V_0^{\pi^e}(s) - \max_a f_0^t(s,a)\right)\right]}_{\Gamma_2} \tag{B.11}$$

$$+ \underbrace{\sum_{t=1}^{T} \mathbb{E}_{s\sim\rho}\left[\mathbb{1}\left\{\mathcal{E}_t\right\}\left(\max_a f_0^t(s,a) - V_0^{\pi^{f^t}}(s)\right)\right]}_{\Gamma_3}.$$

For $\Gamma_0$, we have

$$\Gamma_0 = T_2 \mathbb{E}_{s\sim\rho}\left(V^{\pi^e}(s) - V^{\mathrm{ref}}(s)\right). \tag{B.12}$$

For $\Gamma_2$, we have

$$\Gamma_2 = \sum_{t=1}^{T} \mathbb{E}_{s\sim\rho}\left[\mathbb{1}\left\{\mathcal{E}_t\right\}\left(V_0^{\pi^e}(s) - \max_a f_0^t(s,a)\right)\right] \overset{(i)}{\leq} \sum_{t=1}^{T} \mathbb{1}\left\{\mathcal{E}_t\right\} \sum_{h=0}^{H-1} \mathbb{E}_{s,a\sim d_h^{\pi^e}}\left[\mathcal{T}f_{h+1}^t(s,a) - f_h^t(s,a)\right]$$

$$\overset{(ii)}{\leq} \sum_{t=1}^{T}\left[C_{\pi^e}^{\mathrm{ref}} \cdot \mathbb{1}\left\{\mathcal{E}_t\right\} \sqrt{\sum_{h=0}^{H-1} \mathbb{E}_{s,a\sim\nu_h}\left[\left(f_h^t(s,a) - \mathcal{T}f_{h+1}^t(s,a)\right)^2\right]}\right] \overset{(iii)}{\leq} T_1 C_{\pi^e}^{\mathrm{ref}} \sqrt{H \cdot \Delta_{\mathrm{off}}}, \tag{B.13}$$

where inequality $(i)$ holds because of Lemma G.6, inequality $(ii)$ holds by the definition of $C_{\pi^e}^{\mathrm{ref}}$ (Definition B.4), inequality $(iii)$ holds by Lemma G.5, and Assumption B.3. Note that the telescoping decomposition technique in the above equation

also appears in (Xie & Jiang, 2020; Jin et al., 2021a; Du et al., 2021). Next, we will bound $\Gamma_1 + \Gamma_3$:

$$
\begin{aligned}
\Gamma_1 + \Gamma_3 &= \sum_{t=1}^{T} \left( \mathbb{1}\{\mathcal{E}_t\} + \mathbb{1}\{\bar{\mathcal{E}}_t\} \right) \mathbb{E}_{s \sim d_0} \left[ \max_a f_0^t(s,a) - V_0^{\pi^{f^t}}(s) \right] \\
&\overset{(i)}{\leq} \sum_{t=1}^{T} \left( \mathbb{1}\{\mathcal{E}_t\} + \mathbb{1}\{\bar{\mathcal{E}}_t\} \right) \sum_{h=0}^{H-1} \left| \mathbb{E}_{s,a \sim d_h^{\pi^{f^t}}} \left[ f_h^t(s,a) - \mathcal{T}f_{h+1}^t(s,a) \right] \right| \\
&\overset{(ii)}{=} \sum_{t=1}^{T} \left[ \left( \mathbb{1}\{\mathcal{E}_t\} + \mathbb{1}\{\bar{\mathcal{E}}_t\} \right) \sum_{h=0}^{H-1} \left| \langle X_h(f^t), W_h(f^t) \rangle \right| \right] \\
&\overset{(iii)}{\leq} \sum_{t=1}^{T} \left[ \left( \mathbb{1}\{\mathcal{E}_t\} + \mathbb{1}\{\bar{\mathcal{E}}_t\} \right) \sum_{h=0}^{H-1} \left\| X_h(f^t) \right\|_{\Sigma_{t-1;h}^{-1}} \sqrt{\Delta_{\mathrm{on}} + \lambda B_W^2} \right],
\end{aligned}
\tag{B.14}
$$

where inequality $(i)$ holds by Lemma G.7, equation $(ii)$ holds by the definition of Bilinear model (equation G.2 in Definition G.2), inequality $(ii)$ holds by Lemma G.8. Using Lemma G.9, we have that

$$
\begin{aligned}
\Gamma_1 + \Gamma_3 &\leq \sum_{t=1}^{T} \left[ \left( \mathbb{1}\{\mathcal{E}_t\} + \mathbb{1}\{\bar{\mathcal{E}}_t\} \right) \sum_{h=0}^{H-1} \left\| X_h(f^t) \right\|_{\Sigma_{t-1;h}^{-1}} \sqrt{\Delta_{\mathrm{on}} + \lambda B_W^2} \right] \\
&\overset{(i)}{\leq} H \sqrt{2d \log \left( 1 + \frac{T_1 B_X^2}{\lambda d} \right) \cdot (\Delta_{\mathrm{on}} + \lambda B_W^2) \cdot T_1} + H \sqrt{2d_{\mathrm{ref}} \log \left( 1 + \frac{T_2 B_X^2}{\lambda d_{\mathrm{ref}}} \right) \cdot (\Delta_{\mathrm{on}} + \lambda B_W^2) \cdot T_2} \\
&\overset{(ii)}{\leq} H \left( \sqrt{2d \log \left( 1 + \frac{T_1}{d} \right) \cdot (\Delta_{\mathrm{on}} + B_X^2 B_W^2) \cdot T_1} + \sqrt{2d_{\mathrm{ref}} \log \left( 1 + \frac{T_2}{d_{\mathrm{ref}}} \right) \cdot (\Delta_{\mathrm{on}} + B_X^2 B_W^2) \cdot T_2} \right),
\end{aligned}
\tag{B.15}
$$

where inequality $(i)$ holds by the assumption that $\mathcal{C}_{\mathrm{ref}}$ has bilinear rank $d_{\mathrm{ref}}$, and inequality $(ii)$ holds by plugging in $\lambda = B_X^2$. Substituting equation B.12, inequality B.13, and inequality equation B.15 into equation B.11, we have

$$
\begin{aligned}
\sum_{t=1}^{T} V^{\pi^e} - V^{\pi^{f^t}} &\leq \Gamma_0 + \Gamma_2 + \Gamma_1 + \Gamma_3 \leq T_2 \left( V^{\pi^e}(s) - V^{\mathrm{ref}}(s) \right) + T_1 C_{\pi^e}^{\mathrm{ref}} \sqrt{H \cdot \Delta_{\mathrm{off}}} \\
&\quad + H \left( \sqrt{2d \log \left( 1 + \frac{T_1}{d} \right) \cdot (\Delta_{\mathrm{on}} + B_X^2 B_W^2) \cdot T_1} + \sqrt{2d_{\mathrm{ref}} \log \left( 1 + \frac{T_2}{d_{\mathrm{ref}}} \right) \cdot (\Delta_{\mathrm{on}} + B_X^2 B_W^2) \cdot T_2} \right)
\end{aligned}
\tag{B.16}
$$

Plugging in the values of $\Delta_{\mathrm{on}}, \Delta_{\mathrm{off}}$ from equation G.5 and equation G.6, and using the subadditivity of the square root function, we have

$$
\begin{aligned}
\sum_{t=1}^{T} V^{\pi^e} - V^{\pi^{f^t}} &\leq T_2 \left( V^{\pi^e}(s) - V^{\mathrm{ref}}(s) \right) + 16 V_{\max} C_{\pi^e}^{\mathrm{ref}} T_1 \sqrt{\frac{H}{m_{\mathrm{off}}} \log \left( \frac{2HT_1 |\mathcal{F}|}{\delta} \right)} \\
&\quad + \left( 16 V_{\max} \sqrt{\frac{1}{m_{\mathrm{on}}} \log \left( \frac{2HT_1 |\mathcal{F}|}{\delta} \right)} + B_X B_W \right) \cdot H \sqrt{2dT_1 \log \left( 1 + \frac{T_1}{d} \right)} \\
&\quad + \left( 16 V_{\max} \sqrt{\frac{1}{m_{\mathrm{on}}} \log \left( \frac{2HT_2 |\mathcal{F}|}{\delta} \right)} + B_X B_W \right) \cdot H \sqrt{2d_{\mathrm{ref}} T_2 \log \left( 1 + \frac{T_2}{d_{\mathrm{ref}}} \right)}
\end{aligned}
\tag{B.17}
$$

Setting $m_{\mathrm{off}} = T, m_{\mathrm{on}} = 1$ in the above equation completes the proof, we have

$$
\begin{aligned}
\sum_{t=1}^{T} V^{\pi^e} - V^{\pi^t} &\leq \widetilde{O} \left( C_{\pi^e}^{\mathrm{ref}} \sqrt{HT_1} \right) + \widetilde{O} \left( H\sqrt{dT_1} \right) + T_2 \left( V^{\pi^e}(s) - V^{\mathrm{ref}}(s) \right) + \widetilde{O} \left( H\sqrt{d_{\mathrm{ref}} T_2} \right) \\
&\leq \begin{cases} \widetilde{O} \left( C_{\pi^e}^{\mathrm{ref}} H\sqrt{dT_1} \right) & \text{if } T_1 \gg T_2, \\ \widetilde{O} \left( T_2 \left( V^{\pi^e} - V^{\mathrm{ref}} \right) + H\sqrt{d_{\mathrm{ref}} T_2} \right) & \text{otherwise.} \end{cases} \\
&\leq \widetilde{O} \left( \min \left\{ C_{\pi^e}^{\mathrm{ref}} H\sqrt{dT}, \; T \left( V^{\pi^e} - V^{\mathrm{ref}} \right) + H\sqrt{d_{\mathrm{ref}} T} \right\} \right),
\end{aligned}
\tag{B.18}
$$

where the last inequality holds because $T_1, T_2 \leq T$, which completes the proof. $\qquad \square$

## C. Environment Details

**Antmaze.** The `antmaze` navigation tasks aim to control an 8-DoF ant quadruped robot to move from a starting point to a desired goal in a maze. The agent will receive sparse 0-1 rewards depending on whether it reaches the goal or not. We study each method on "medium" and "hard" (shown in Figure 4) mazes which are difficult to solve, using the following datasets from D4RL (Fu et al., 2020): `large-diverse`, `large-play`, `medium-diverse`, and `medium-play`. The difference between "diverse" and "play" datasets is the optimality of the trajectories they contain. The "diverse" datasets contain the trajectories commanded to a random goal from random starting points, while the "play" datasets contain the trajectories commanded to specific locations which are not necessarily the goal. For Cal-QL, CQL and IQL, we pre-trained the agent using the offline dataset for 1M steps. For the online learning phase, each method was trained for 1M environment steps, taking 1 update per environment step.

**Franka Kitchen.** The `kitchen` tasks require controlling a 9-DoF Franka robot to arrange a kitchen environment into a desired configuration. The configuration is decomposed into 4 subtasks, and the agent will receive rewards of 0, $+1$, $+2$, $+3$, or $+4$ depending on how many subtasks it has managed to solve. To solve the whole task and reach the desired configuration, it is important to learn not only how to solve each subtask, but also to figure out the correct order to solve. We study this domain using datasets with three different optimalities: `kitchen-complete`, `kitchen-partial`, and `kitchen-mixed`. The "complete" dataset contains the trajectories of the robot performing the whole task completely. The "partial" dataset partially contains some complete demonstrations, while others are incomplete demonstrations solving the subtasks. The "mixed" dataset only contains incomplete data without any complete demonstrations, which is hard and requires the highest degree of stitching and generalization ability. For Cal-QL, CQL, and IQL, we pre-trained the agent using the offline dataset for 500K steps. Each method was then trained for a total of 1.25M environment steps on each of the 3 kitchen tasks, taking 1 update per environment step.

**Adroit.** The Adroit domain involves controlling a 24-DoF shadow hand robot. There are 3 tasks we consider in this domain: `pen-binary`, `relocate-binary`, `relocate-binary`. These tasks comprise a limited set of narrow human expert data distributions ($\sim 25$) with additional trajectories collected by a behavior-cloned policy. We used the positive segments of each trajectory (when the positive reward signal is found) for all methods. This domain has a very narrow dataset distribution and a large action space. In addition, learning in this domain is made more difficult due to the sparse reward formulation, which leads to exploration challenges. We utilized a variant of the dataset used in prior work (Nair et al., 2020a) to have a standard comparison with SOTA offline finetuning experiments that consider this domain. For the offline learning phase, we pre-trained the agent for 20K steps. For the online phase, the `door-binary` and `relocate-binary` tasks were trained for 1M environment steps for each method. For the `pen-binary` task, each method was trained for 200k steps. We take 1 update per environment step.

**Robotic Manipulation Domain.** The Robotic Manipulation domain consists of a pick-and-place task. This task is a multitask formulation explored in the work, Pretraining for Robots (PTR) (Kumar et al., 2022). Here each task is defined as placing an object in a bin. A distractor object was present in the scene as an adversarial object which the agent had to avoid picking. There were 10 unique objects and no overlap between the task objects and the interfering/distractor objects. For the offline phase, we pre-trained the policy with offline data for 50K steps. For the online phase, there were 40K environment steps taken for each of the methods in the visual manipulation domain. Here we take 5 updates per environment step.

## D. Experiment Details

### D.1. Normalized Scores for all Tasks

The `visual-manipulation`, `adroit`, and `antmaze` domains are all goal-oriented, sparse reward tasks. In these domains, we computed the normalized metric as simply the goal achieved rate for each method. For example, in the visual manipulation environment, if the object was placed successfully in the bin, a $+1$ reward was given to the agent and the task is completed. Similarly, for the `door-binary` task in the adroit tasks, we considered the success rate of opening the door.

For the `kitchen` task, the task is to solve a series of 4 sub-tasks that need to be solved in an iterative manner. The normalized score is computed simply as $\frac{\#\text{tasks solved}}{\text{total tasks}}$.

### D.2. Mixing Ratio Hyperparameter

In this work, we explore the mixing ratio $m$. The mixing ratio is either a value in the range $[0, 1]$ or the value -1. Note that in our formulation, you are able to see the experience from offline trajectories to ground learning and prevent forgetfulness of your pre-trained representation. If this mixing ratio is within $[0, 1]$, it represents what percentage of offline and online data is seen in each batch when fine-tuning. Each element of the batch independently is chosen to be from the offline or online dataset with weight $m$. This constructed batch is used to train the agent. Instead, if the mixing ratio is -1, the buffers are appended to each other and sampled uniformly.

### D.3. Update to Data Ratio, UTD

In the `visual-manipulation` task, we also explored the update to data ratio (UTD) for each approach to see if having more gradient steps per environment step would aid learning by requiring fewer samples. We see that for Cal-QL, UTD 5 is more sample efficient than UTD 1 and thus included that result in the main paper. However, for alternative approaches such as IQL, we don't see any change in the sample efficiency when learning with a different UTD ratio, which is also discussed in Section E. In Figure 8, you could see the performance for both UTD 1 and 5 for Cal-QL and baseline methods.

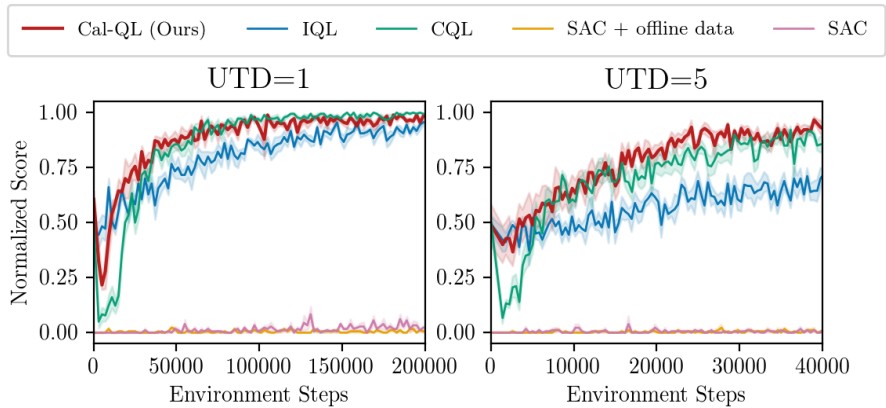

Figure 8. **UTD Abalation**: We show how the change in the update to target ratio can lead to higher sample efficiency for Cal-QL but lead to similar sample efficiency for alternative approaches.

### D.4. Hyperparameters for CQL and Cal-QL

For the domains we tested on, our architecture and learning rate design choices matched for Cal-QL and CQL (Kumar et al., 2020). Below, we will describe some of the design choices we considered and ablated over for these two method types.

**Adroit domains.** In the Adroit tasks, we ablated over the value of the penalty of the CQL regularizer $\mathcal{R}(\theta)$ with the dual version of CQL. The network architecture was a 3-hidden layer for the critic and a 2-hidden layer for the actor networks with a hidden dimension size of 512. A critic learning rate of 3e-4 and an actor learning rate of 1e-4 were used. We utilized a variant of Bellman backup that computes the target value by performing a maximization over target values computed for $k = 10$ actions sampled from the policy at the next state. In the table below, we describe hyperparameters that were swept over for the method(s) across 6 seeds.

Table 3. CQL, Cal-QL Adroit Hyperparameters

| Hyperparameters | Values |
| --- | --- |
| offline $\alpha$ | 0.1, 1, 5, 10, 20 |
| online $\alpha$ | 0.1, 1, 5, 10, 20 |
| mixing ratio | 0, 0.25, 0.5 |

**Antmaze domains.** In the antmaze tasks, we ablated over the value of the threshold of the CQL regularizer $\mathcal{R}(\theta)$ with the dual version of CQL. We used 4-hidden layer critic and 2-hidden layer actor networks with layers of size 256, a critic learning rate of 3e-4, and an actor learning rate of 1e-4. We utilized the Bellman backup that computes the target value by performing a maximization over target values computed for $k = 10$ actions sampled from the policy at the next state. We ablated over the following remaining set of Hyperparameters:

*Table 4.* CQL, Cal-QL Antmaze Hyperparameters

| Hyperparameters | Values |
|---|---|
| offline $\tau$ | 0.1, 0.4, 0.8 |
| online $\tau$ | 0.1, 0.4, 0.8 |
| mixing ratio | 0.5 |

**Visual Robotic Domains.** For the visual pick and place domains, we follow exactly the same hyperparameters as the CQL implementation from COG (Singh et al., 2020): a critic learning rate of 3e-4, an actor learning rate of 1e-4, using $k = 4$ actions from the policy for computing the target values for computing the TD error, and using $k = 4$ actions to compute the log-sum-exp in CQL. We ablated over the following remaining set of Hyperparameters:

*Table 5.* CQL/CalQL Robotic Hyperparameters

| Hyperparameters | Values |
|---|---|
| offline $\alpha$ | 1, 10 |
| online $\alpha$ | 0.1, 0.5, 1, 2, 5, 10, 20 |
| mixing ratio | 0, 0.25, 0.5 |

### D.4.1. HYPERPARAMETERS FOR IQL

**Adroit domains.** In the Adroit tasks, we ablated over the value of the expectile $\tau$ and temperature $\beta$. The network architecture was a 2-hidden layer that was used for both the critic and actor networks with a hidden dimension size of 256. A critic learning rate of 3e-4 and an actor learning rate of 1e-4 were used. In the table below, we describe hyperparameters that were swept over for the method(s) across 6 seeds. For this domain, we utilized the author's recommended parameters which they swept and abalated over in their work. Below are those chosen parameters and others we swept over.

*Table 6.* IQL Adroit Hyperparameters

| Hyperparameters | Values |
|---|---|
| expectile $\tau$ | 0.7 |
| temperature $\beta$ | 3.0 |
| mixing ratio | 0, 0.25, 0.5, -1 |

**Antmaze domains.** In the AntMaze tasks, we ablated over the value of the expectile $\tau$ and temperature $\beta$. Following the original IQL paper, we used author suggested hyperparameters: 2-hidden layer critic and actor networks with layers of size 256, a critic learning rate of 3e-4 and an actor learning rate of 1e-4. The others are shown below:

*Table 7.* IQL Antmaze Hyperparameters

| Hyperparameters | Values |
|---|---|
| expectile $\tau$ | 0.9 |
| temperature $\beta$ | 10 |
| mixing ratio | 0, 0.25, 0.5, -1 |

**Visual Robotic Domains.** For the visual pick and place domains, we follow exactly the same hyperparameters for IQL with respect to architecture and hyperparameter choices as CQL. This includes a critic learning rate of 3e-4, and actor learning rate of 1e-4, with the ConvNet architecture choice from COG (Singh et al., 2020). We ablated over the following remaining set of hyperparameters:

*Table 8.* IQL Robotic Hyperparameters

| Hyperparameters | Values |
|---|---|
| expectile $\tau$ | 0.8, 0.9, 0.95, 0.99 |
| temperature $\beta$ | 1, 2, 3, 5, 10, 25, 50, 100 |
| mixing ratio | 0, 0.25, 0.5, -1 |

Below, we will show an ablation study detailing how the tuning of the temperature $\beta$ value in the offline and online phases leads to changes in performance. In Figure 9, we see that we have extensively swept over several temperature values to choose the optimal checkpoint for downstream evaluation.

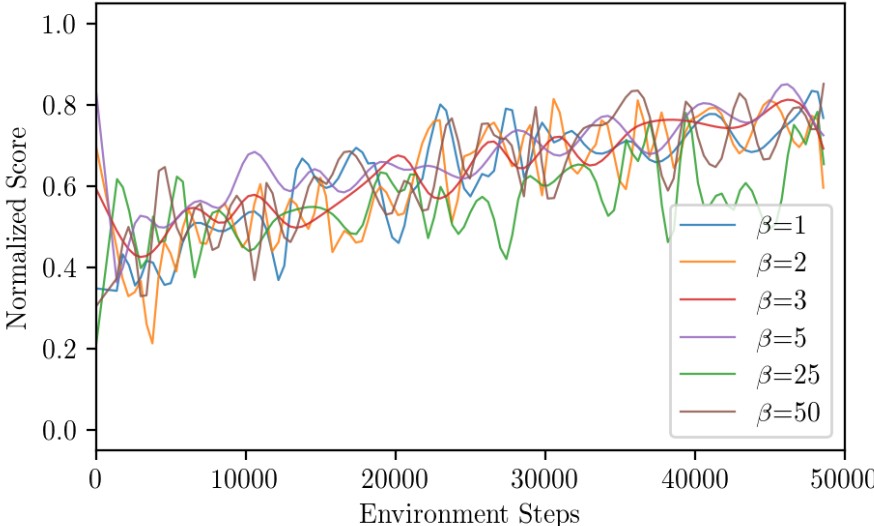

*Figure 9.* **IQL Abalation**: We show how the change in the policy temperature $\beta$ has little to no effect on the ultimate policy performance.

### D.4.2. HYPERPARAMETERS FOR SAC, SAC + ONLINE DATA

We use the standard hyperparameters for SAC as derived from the original implementation in (Haarnoja et al., 2018a). We used the same network architecture choice (including hyperparameters) as CQL. We used automatic entropy tuning for the policy and critic entropy terms, with a target entropy of the negative action dimension.

## E. Extended Discussion on Limitations of Existing Fine-Tuning Methods

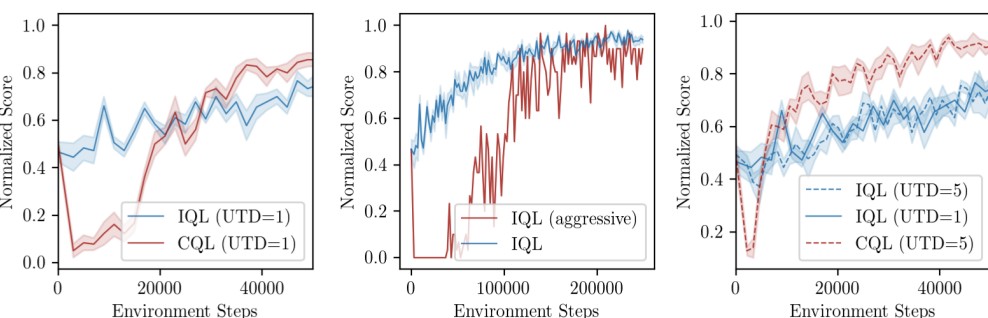

*Figure 10.* **IQL and CQL:** Step 0 on the x-axis is the performance after offline pre-training. Observe while CQL suffers from initial policy unlearning, IQL improves slower throughout fine-tuning.

In this section, we aim to highlight some potential reasons behind the slow improvement of IQL in our empirical analysis experiment in Section 4.1. A natural hypothesis is that IQL improves slowly because we are not making enough updates per unit of data collected by the environment. To investigate this, we ran IQL with **(a)** five times as many gradient steps per step of data collection, and **(b)** with a more aggressive policy update. Observe in Figure 10 that **(a)** does not improve the asymptotic performance of IQL, although it does improve CQL meaning that there is room for improvement on this task by making more gradient updates. Observe in Figure 10 that **(b)** often induces to policy unlearning, similar to the failure mode in CQL. These two observations together indicate that a policy constraint approach can slow down learning asymptotically, and we cannot increase the speed by making more aggressive updates as this causes the policy to find erroneously optimistic out-of-distribution actions, and unlearn the policy learned from offline data. An alternative would be for our method to altogether avoid policy constraints.

## F. Discussion of Policy Unlearning in Fine-Tuning

### F.1. Bellman Consistency Equation of CQL

To analyze policy unlearning with conservative methods at the beginning of online fine-tuning, we consider a tabular setting, where we are learning conservative value functions using a generic policy-iteration style offline RL method based on CQL (Kumar et al., 2020). Our goal is to understand the differences in a policy obtained by running one round of policy improvement with and without additional online data. Key to our analysis is the Bellman backup induced by CQL (Equation 3.1):

$$Q^\pi(s,a) = (\mathcal{B}^\pi Q^\pi)(s,a) - \alpha \left[ \frac{\pi(a|s)}{\pi_\beta(a|s)} - 1 \right]. \tag{F.1}$$

By expanding $\mathcal{B}^\pi$, Equation F.1 can also be interpreted as running standard Q-iteration in an MDP with a pessimistic reward function, which depends upon the learned policy $\pi$, the behavior policy $\pi_\beta$ induced by the dataset, and the coefficient $\alpha$ from Equation 3.1: $r^\pi_{\alpha,\beta}(s,a) = r(s,a) - \alpha \left[ \frac{\pi(a|s)}{\pi_\beta(a|s)} - 1 \right]$. This means that once online fine-tuning commences with a new regularizer $\alpha$, and a new behavior policy $\pi_{\beta'}$ induced by online data added to the buffer, the pessimistic reward function $r^\pi_{\alpha,\beta}$ may *bias towards* $r^\pi_{\alpha',\beta'}$. Hence, the policy improvement on the resulting Q-function with the online data may simply not lead to any policy improvement on the ground-truth reward function. We will first provide a condition such that the pessimistic Q-function is invariant to such reward bias during the online fine-tuning, and then show how our approach alleviates the reward bias.

**Performance difference during fine-tuning.** Consider the fixed points from solving Equation F.1 w.r.t. the biased rewards $r^\pi_{\alpha,\beta}$ and $r^\pi_{\alpha',\beta'}$, Theorem F.1 provides a necessary and sufficient condition for the fixed points to be invariant to reward bias.

**Theorem F.1** (Invariant Conservative Q Functions). *Let $Q$ and $Q'$ denote the conservative value function from solving the fixed point Equation F.1 with regularizers $\alpha, \alpha'$ and behavior policies $\pi_\beta, \pi_{\beta'}$ respectively. Then for a given policy $\pi$,*

$Q(s, a) = Q'(s, a), \forall s \in \mathcal{S}$ *if and only if*

$$\frac{\alpha}{\pi_\beta(a|s)} - \frac{\alpha'}{\pi_{\beta'}(a|s)} = \frac{\alpha - \alpha'}{\pi(a|s)}, \ \forall (s, a) \in \mathcal{S} \times \mathcal{A}. \tag{F.2}$$

The proof of Theorem F.1 is provided in Appendix F.5. Theorem F.1 implies that one shall expect a performance change in the fine-tuning phase, whenever Equation F.2 becomes invalid. Unfortunately, Equation F.2 is generally hard to enforce in practice since the new behavior policy $\pi_{\beta'}$ and the updated policy $\pi$ may be intractable, as suggested by the performance dip in Section 4.1.

**Preventing unlearning via calibration.** Since the standard CQL training objective suffers from reward bias, which may continue to deteriorate during fine-tuning. The next corollary provides a condition under which $r^\pi_{\alpha,\beta}(s, a)$ becomes unbiased.

**Corollary F.2.** *The reward function* $r^\pi_{\alpha,\beta}(s, a)$ *induced by the CQL training (Equation F.1) becomes unbiased when* $\pi = \pi_\beta$:
$r^{\pi_\beta}_{\alpha,\beta}(s, a) = r(s, a), \forall (s, a) \in \mathcal{S} \times \mathcal{A}$.

The proof of Corollary F.2 is straight forward as substituting $\pi = \pi_\beta$ will set $\left[\frac{\pi(a|s)}{\pi_\beta(a|s)} - 1\right] = 0$, which makes Equation F.1 become a Bellman backup w.r.t. the original reward $r(s, a)$. Corollary F.2 implies that the CQL-induced Bellman backup (Equation 3.1) becomes unbiased when the updating policy equals the behavior policy. Hence, if one aims at using a neural network $Q_\theta$ to approximate the CQL Bellman backup w.r.t. behavior policy $\pi_\beta$, one shall consider using $Q^{\pi_\beta}(s, a)$ to calibrate the bias. This empirically implies that we shall use $Q^{\pi_\beta}(s, a)$ and $Q^{\pi_{\beta'}}(s, a)$ to calibrate the $Q_\theta$ during offline and online phase, as we have presented in Definition 4.1 and Section 5.

## F.2. Notations

In this subsection, we provide the notations for deriving the Bellman Consistency Equation of the conservative Bellman Consistency equation (CQL Objective) equation F.1. Our matrix notation follows (Li et al., 2020).

- We consider the infinite horizon tabular setting where $\mathcal{S}$ and $\mathcal{A}$ are discrete and finite, $\gamma \in (0, 1)$ is a discount factor and $r : \mathcal{S} \times \mathcal{A} \mapsto [0, 1]$ is the reward function;

- The value function $V^\pi(s)$ of a state w.r.t. policy $\pi$ is defined as

$$V^\pi(s) := \mathbb{E}\left[\sum_{t=0}^\infty \gamma^t r(s^t, a^t)|s^0 = s\right], \ \forall s \in \mathcal{S}; \tag{F.3}$$

- The Q function $Q^\pi(s, a)$ of a state action pair $(s, a)$ w.r.t. a policy $\pi$ is defined by

$$Q^\pi(s, a) := \mathbb{E}\left[\sum_{t=0}^\infty \gamma^t r(s^t, a^t)|s^0 = s, a^0 = a\right] \forall (s, a) \in \mathcal{S} \times \mathcal{A}; \tag{F.4}$$

- $\mathbf{P} \in \mathbb{R}^{|\mathcal{S}||\mathcal{A}| \times |\mathcal{S}|}$ is a matrix of the transition kernel $P$;

- $\mathbf{P}^\pi \in \mathbb{R}^{|\mathcal{S}||\mathcal{A}| \times |\mathcal{S}||\mathcal{A}|}$ and $\mathbf{P}_\pi \in \mathbb{R}^{|\mathcal{S}| \times |\mathcal{S}|}$ two square probability transition matrices induced by the policy $\pi$ over the state-action pair and the states respectively, defined by

$$\mathbf{P}^\pi := \mathbf{P}\mathbf{\Pi}^\pi, \quad \mathbf{P}_\pi := \mathbf{\Pi}^\pi \mathbf{P}; \tag{F.5}$$

- $\mathbf{\Pi}^\pi \in \{0, 1\}^{|\mathcal{S}| \times |\mathcal{S}||\mathcal{A}|}$ is a projection matrix:

$$\mathbf{\Pi}^\pi = \begin{pmatrix} \mathbf{e}^\top_{\pi(1)} & & & \\ & \mathbf{e}^\top_{\pi(2)} & & \\ & & \ddots & \\ & & & \mathbf{e}^\top_{\pi(|\mathcal{S}|)} \end{pmatrix}. \tag{F.6}$$

- $\mathbf{r} \in \mathbb{R}^{|\mathcal{S}||\mathcal{A}|}$ is the reward function

- $\mathbf{r}^\pi \in \mathbb{R}^{|\mathcal{S}|}$ is the reward function following policy $\pi$, simply we have $\mathbf{r}^\pi = \mathbf{\Pi}^\pi \mathbf{r}$.

### F.3. Derivations of Conservative Bellman Consistency Equation F.1

$\forall (s,a) \in \mathcal{S} \times \mathcal{A}$, the tabular CQL optimization (Equation 3.1) aims at solving the following optimization problem:

$$\min_{Q(s,a)} \alpha \left[ \mathop{\mathbb{E}}_{s \sim \mathcal{D}, a \sim \pi} Q(s,a) - \mathop{\mathbb{E}}_{(s,a) \sim \mathcal{D}, a \sim \pi_\beta} Q(s,a) \right] + \frac{1}{2} \mathop{\mathbb{E}}_{s \sim \mathcal{D}, a \sim \pi_\beta} \left[ (Q(s,a) - \mathcal{B}^\pi Q(s,a))^2 \right]. \tag{F.7}$$

If we rewrite $x = Q(s,a)$ and define function $f(x)$ as

$$f(x) := \alpha \left[ \mathop{\mathbb{E}}_{s \sim \mathcal{D}, a \sim \pi} x - \mathop{\mathbb{E}}_{(s,a) \sim \mathcal{D}, a \sim \pi_\beta} x \right] + \frac{1}{2} \mathop{\mathbb{E}}_{s \sim \mathcal{D}, a \sim \pi_\beta} \left[ (x - \mathcal{B}^\pi x)^2 \right], \tag{F.8}$$

setting $f'(x) = 0$ yields

$$\alpha \left( \pi(a|s) - \pi_\beta(a|s) \right) + \pi_\beta (x - \mathcal{B}^\pi x) = 0, \tag{F.9}$$

which leads to

$$x = \mathcal{B}^\pi x - \alpha \left[ \frac{\pi(a|s)}{\pi_\beta(a|s)} - 1 \right]. \tag{F.10}$$

### F.4. Conservative Bellman Consistency Equations

**Q Functions.** Considering the matrix form and point-wise Bellman consistency equation, we have

$$\mathbf{Q}^\pi = \mathbf{r} + \gamma \mathbf{P}^\pi \mathbf{Q}^\pi \implies \mathbf{Q}^\pi = (\mathbf{I} - \gamma \mathbf{P}^\pi)^{-1} \mathbf{r} \tag{F.11}$$

$$Q^\pi(s,a) = r(s,a) + \gamma \left[ \mathbf{P}^\pi \mathbf{Q}^\pi \right]_{(s,a)}, \forall (s,a) \in \mathcal{S} \times \mathcal{A}, \tag{F.12}$$

where we use $\left[ \mathbf{P}^\pi \mathbf{Q}^\pi \right]_{(s,a)}$ to denote the entries $(s,a)^{\text{th}}$ of the matrix $\mathbf{P}^\pi \mathbf{Q}^\pi$. Now recall the point-wise Bellman consistency equation of the CQL objective equation F.1, we also have the following point-wise consistency equation:

$$Q_{\alpha,\beta}^\pi(s,a) = r(s,a) + \gamma \left[ \mathbf{P}^\pi \mathbf{Q}_{\alpha,\beta}^\pi \right]_{(s,a)} - \alpha \left[ \frac{\pi(a|s)}{\pi_\beta(a|s)} - 1 \right], \forall (s,a) \in \mathcal{S} \times \mathcal{A}, \tag{F.13}$$

$$= r_{\alpha,\beta} + \gamma \left[ \mathbf{P}^\pi \mathbf{Q}^\pi \right]_{(s,a)}, \tag{F.14}$$

where

$$r_{\alpha,\beta}^\pi(s,a) := r(s,a) - \alpha \left[ \frac{\pi(a|s)}{\pi_\beta(a|s)} - 1 \right], \forall (s,a) \in \mathcal{S} \times \mathcal{A}. \tag{F.15}$$

Hence, we can similarly have the bellman-consistency equation of CQL in matrix form:

$$\mathbf{Q}_{\alpha,\beta} = \mathbf{r}_{\alpha,\beta} + \gamma \mathbf{P}^\pi \mathbf{Q}_{\alpha,\beta}^\pi \implies \mathbf{Q}_{\alpha,\beta}^\pi = (\mathbf{I} - \gamma \mathbf{P}^\pi)^{-1} \mathbf{r}_{\alpha,\beta}. \tag{F.16}$$

**Value Functions.** Now considering the Bellman Consistency equation of the Value function, we have

$$\mathbf{V}^\pi = \mathbf{r}^\pi + \gamma \mathbf{P}_\pi \mathbf{V}^\pi \implies \mathbf{V}^\pi = (\mathbf{I} - \gamma \mathbf{P}_\pi)^{-1} \mathbf{r}^\pi \tag{F.17}$$

$$\mathbf{V}_{\alpha,\beta}^\pi = \mathbf{r}_{\alpha,\beta}^\pi + \gamma \mathbf{P}_\pi \implies \mathbf{V}_{\alpha,\beta}^\pi = (\mathbf{I} - \gamma \mathbf{P}_\pi)^{-1} \mathbf{r}_{\alpha,\beta}^\pi. \tag{F.18}$$

**Summary.** In summary, for a given reward function $\mathbf{r}$, a fixed policy $\pi$, a behavior policy $\pi_\beta$, and a fixed constant $\alpha$, the *policy evaluation* for CQL satisfies:

$$\begin{aligned} \mathbf{V}_{\alpha,\beta}^\pi &= (\mathbf{I} - \gamma \mathbf{P}_\pi)^{-1} \mathbf{r}_{\alpha,\beta}^\pi = (\mathbf{I} - \gamma \mathbf{P}_\pi)^{-1} \mathbf{\Pi}^\pi \left[ \mathbf{r} - \alpha \left( \pi/\pi_\beta - 1 \right) \right] \\ \mathbf{Q}_{\alpha,\beta}^\pi &= (\mathbf{I} - \gamma \mathbf{P}^\pi)^{-1} \mathbf{r}_{\alpha,\beta} = (\mathbf{I} - \gamma \mathbf{P}^\pi)^{-1} \left[ \mathbf{r} - \alpha \left( \pi/\pi_\beta - 1 \right) \right]. \end{aligned} \tag{F.19}$$

where $\pi/\pi_\beta \in \mathbb{R}^{|\mathcal{S}| \times |\mathcal{A}|}$ is a vector whose $(s,a)$ entry denotes $\pi(a|s)/\pi_\beta(a|s)$ and $\mathbf{1} = \{1, 1, \ldots, 1\}^\top \in \mathbb{R}^{|\mathcal{S}||\mathcal{A}|}$.

## F.5. Proof of Theorem F.1

**Theorem F.3** (Invariant Conservative Q Functions). *Let $Q^\pi_{\alpha,\beta}$ and $Q^\pi_{\alpha',\beta'}$ denote the conservative value function from solving the conservative bellman consistency equation (Equation F.1 and F.19) with regularizers $\alpha, \alpha'$ and behavior policies $\pi_\beta, \pi_{\beta'}$ respectively. Then for a given policy $\pi$, $Q^\pi_{\alpha,\beta}(s) = Q^\pi_{\alpha',\beta'}(s), \forall s \in \mathcal{S}$ if and only if*

$$\frac{\alpha}{\pi_\beta(a|s)} - \frac{\alpha'}{\pi_{\beta'}(a|s)} = \frac{\alpha - \alpha'}{\pi(a|s)}, \ \forall (s,a) \in \mathcal{S} \times \mathcal{A}. \tag{F.20}$$

*Proof.* By the conservative Bellman Consistency equation F.19, we know that changing a behavior policy from $\pi_\beta$ to $\pi_{\beta'}$ and changing the regularize from $\alpha$ to $\alpha'$, we have

$$\begin{aligned}
\mathbf{Q}^\pi_{\alpha,\beta} - \mathbf{Q}^\pi_{\alpha',\beta'} &= (\mathbf{I} - \gamma \mathbf{P}^\pi)^{-1}(\mathbf{r}^\pi_{\alpha,\beta} - \mathbf{r}^\pi_{\alpha',\beta'}) \\
&= (\mathbf{I} - \gamma \mathbf{P}^\pi)^{-1}\left[\alpha\left(\frac{\pi}{\pi_\beta} - \mathbf{1}\right) - \alpha'\left(\frac{\pi}{\pi_{\beta'}} - \mathbf{1}\right)\right].
\end{aligned} \tag{F.21}$$

Since $(\mathbf{I} - \gamma \mathbf{P})^{-1}$ is a square and full rank matrix, $\mathbf{Q}^\pi_{\alpha,\beta} - \mathbf{Q}^\pi_{\alpha',\beta'} = \mathbf{0}$ holds if and only if

$$\alpha\left(\frac{\pi}{\pi_\beta} - \mathbf{1}\right) - \alpha'\left(\frac{\pi}{\pi_{\beta'}} - \mathbf{1}\right) = \mathbf{0} \implies \frac{\alpha}{\pi_\beta(a|s)} - \frac{\alpha'}{\pi_{\beta'}(a|s)} = \frac{\alpha - \alpha'}{\pi(a|s)}, \ \forall (s,a) \in \mathcal{S} \times \mathcal{A}, \tag{F.22}$$

which finishes the proof. $\square$

# G. Key Results of HyQ (Song et al., 2022)

In this section, we restate the major theoretical results of Hy-Q (Song et al., 2022) for completeness.

---

**Algorithm 3** Hybrid Q-learning using offline and online data (Song et al., 2022)

---

1: **Input:** Value function class: $\mathcal{F}$, # iterations: $T$, offline dataset $\mathcal{D}^\nu_h$ of size $m_{\text{off}}$ for $h \in [H-1]$.
2: Initialize $f^1_h(s,a) = 0, \forall(s,a)$.
3: **for** $t = 1, \ldots, T$ **do**
4:     Let $\pi^t$ be the greedy policy w.r.t. $f^t$                                   ▷ I.e.,$\pi^t_h(s) = \arg\max_a f^t_h(s,a)$.
5:     For each $h$, collect $m_{\text{on}}$ online tuples $\mathcal{D}^t_h \sim d^{\pi^t}_h$                  ▷ online data collection
6:     Set $f^{t+1}_H(s,a) = 0, \forall(s,a)$.
7:     **for** $h = H-1, \ldots 0$ **do**                                         ▷ FQI with offline and online data
8:         Estimate $f^{t+1}_h$ using least squares regression on the aggregated data:

$$f^{t+1}_h \leftarrow \arg\min_{f \in \mathcal{F}_h} \left\{ \widehat{\mathbb{E}}_{\mathcal{D}^\nu_h}\left[f(s,a) - r - \max_{a'} f^{t+1}_{h+1}(s',a')\right]^2 + \sum_{\tau=1}^t \widehat{\mathbb{E}}_{\mathcal{D}^\tau_h}\left[f(s,a) - r - \max_{a'} f^{t+1}_{h+1}(s',a')\right]^2 \right\} \tag{G.1}$$

9:     **end for**
10: **end for**
11: **Output:** $\pi^T$

---

## G.1. Assumptions

**Assumption G.1** (Realizability and Bellman completeness). *For any $h$, we have $Q^\star_h \in \mathcal{F}_h$, and additionally, for any $f_{h+1} \in \mathcal{F}_{h+1}$, we have $\mathcal{T} f_{h+1} \in \mathcal{F}_h$.*

**Definition G.2** (Bilinear model Du et al. (2021)). *We say that the MDP together with the function class $\mathcal{F}$ is a bilinear model of rand d of for any $h \in [H-1]$, there exist two (known) mappings $X_h, W_h : \mathcal{F} \mapsto \mathbb{R}^d$ with $\max_f \|X_h(f)\|_2 \le B_X$ and $\max_f \|W_h(f)\|_2 \le B_W$ such that*

$$\forall f, g \in \mathcal{F} : \ \left|\mathbb{E}_{s,a \sim d^{\pi^f}_h}[g_h(s,a) - \mathcal{T} g_{h+1}(s,a)]\right| = |\langle X_h(f), W_h(g)\rangle|. \tag{G.2}$$

**Definition G.3** (Bellman error transfer coefficient). *For any policy $\pi$, we define the transfer coefficient as*

$$C_\pi := \max \left\{ 0, \max_{f \in \mathcal{F}} \frac{\sum_{h=0}^{H-1} \mathbb{E}_{s,a \sim d_h^\pi}[\mathcal{T}f_{h+1}(s,a) - f_h(s,a)]}{\sqrt{\sum_{h=0}^{H-1} \mathbb{E}_{s,a \sim \nu_h}(\mathcal{T}f_{h+1}(s,a) - f_h(s,a))^2}} \right\}. \tag{G.3}$$

## G.2. Main Theorem of Hy-Q

**Theorem G.4** (Theorem 1 of Song et al. (2022)). *Fix $\delta \in (0,1), m_{\text{off}} = T, m_{\text{on}} = 1$, and suppose that the underlying MDP admits Bilinear rank $d$ (Definition G.2), and the function class $\mathcal{F}$ satisfies Assumption G.1. Then with probability at least $1 - \delta$, Algorithm 3 obtains the following bound on cumulative suboptimality w.r.t. any comparator policy $\pi^e$:*

$$\sum_{t=1}^{T} V^{\pi^e} - V^{\pi^t} = \widetilde{O}\left(\max\{C_{\pi^e}, 1\} V_{\max} B_X B_W \sqrt{dH^2T \cdot \log(|\mathcal{F}|/\delta)}\right). \tag{G.4}$$

## G.3. Key Lemmas

### G.3.1. LEAST SQUARES GENERALIZATION AND APPLICATIONS

**Lemma G.5** (Lemma 7 of Song et al. (2022), Online and Offline Bellman Error Bound for FQI). *Let $\delta \in (0,1)$ and $\forall h \in [H-1], t \in [T]$, let $f_h^{t+1}$ be the estimated value function for time step $h$ computed ia least square regression using samples in the dataset $\{\mathcal{D}_h^\nu, \mathcal{D}_h^1, \ldots, \mathcal{D}_h^t\}$ in equation G.1 in the iteration $t$ of Algorithm 3. Then with probability at least $1 - \delta$, for any $h \in [H-1]$ and $t \in [T]$, we have*

$$\left\|f_h^{t+1} - \mathcal{T}f_{h+1}^{t+1}\right\|_{2,\nu_h}^2 \leq \frac{1}{m_{\text{off}}} 256 V_{\max}^2 \log(2HT|\mathcal{F}|/\delta) =: \Delta_{\text{off}} \tag{G.5}$$

*and*

$$\sum_{\tau=1}^{t} \left\|f_h^{t+1} - \mathcal{T}f_{h+1}^{t+1}\right\|_{2,\mu_h^\tau}^2 \leq \frac{1}{m_{\text{on}}} 256 V_{\max}^2 \log(2HT|\mathcal{F}|/\delta) =: \Delta_{\text{on}}, \tag{G.6}$$

*where $\nu_h$ denotes the offline data distribution at time $h$, and the distribution $\mu_h^\tau \in \Delta(s,a)$ is defined such that $s, a \sim d_h^{\pi^\tau}$.*

### G.3.2. BOUNDING OFFLINE SUBOPTIMALITY VIA PERFORMANCE DIFFERENCE LEMMA

**Lemma G.6** (Lemma 5 of Song et al. (2022), performance difference lemma of w.r.t. $\pi^e$). *Let $\pi^e = (\pi_0^e, \ldots, \pi_{H-1}^e)$ be a comparator policy and consider any value function $f = (f_0, \ldots, f_{H-1})$, where $f_h : \mathcal{S} \times \mathcal{A} \mapsto \mathbb{R}$. Then we have*

$$\mathbb{E}_{s \sim d_0} \left[V_0^{\pi^e}(s) - \max_a f_0(s,a)\right] \leq \sum_{i=1}^{H-1} \mathbb{E}_{s,a \sim d_i^{\pi^e}} \left[\mathcal{T}f_{i+1}(s,a) - f_i(s,a)\right], \tag{G.7}$$

*where we define $f_H(s,a) = 0, \forall(s,a)$.*

### G.3.3. BOUNDING ONLINE SUBOPTIMALITY VIA PERFORMANCE DIFFERENCE LEMMA

**Lemma G.7** (Lemma 4 of Song et al. (2022), performance difference lemma). *For any function $f = (f_0, \ldots, f_{H-1})$ where $f_h : \mathcal{S} \times \mathcal{A} \mapsto \mathbb{R}$ and $h \in [H-1]$, we have*

$$\mathbb{E}_{s \sim d_0} \left[\max_a f_0(s,a) - V_0^{\pi^f}(s)\right] \leq \sum_{h=0}^{H-1} \left|\mathbb{E}_{s,a \sim d_h^{\pi^f}}\left[f_h(s,a) - \mathcal{T}f_{h+1}(s,a)\right]\right|, \tag{G.8}$$

*where we define $f_H(s,a) = 0, \forall s, a$.*

**Lemma G.8** (Lemma 8 of Song et al. (2022), upper bounding bilinear class). *For any $t \geq 2$ and $h \in [H-1]$, we have*

$$\left|\langle W_h(f^t), X_h(f^t)\rangle\right| \leq \left\|X_h(f^t)\right\|_{\Sigma_{t-1;h}^{-1}} \sqrt{\sum_{i=1}^{t-1} \mathbb{E}_{s,a \sim d_h^{f^i}}\left[\left(f_h^t - \mathcal{T}f_{h+1}^t\right)^2\right] + \lambda B_W^2}, \tag{G.9}$$

*where $\Sigma_{t-1;h}$ is defined as equation B.5 and we use $d_h^{f^i}$ to denote $d_h^{\pi^{f^i}}$.*

**Lemma G.9** (Lemma 6 of Song et al. (2022), bounding the inverse covariance norm). *Let $X_h(f^1), \ldots, X_h(f^T) \in \mathbb{R}^d$ be a sequence of vectors with $\|X_h(f^t)\|_2 \leq B_X < \infty, \forall t \leq T$. Then we have*

$$\sum_{t=1}^{T} \|X_h(f^t)\|_{\Sigma_{t-1;h}^{-1}} \leq \sqrt{2dT \log\left(1 + \frac{TB_X^2}{\lambda d}\right)}, \tag{G.10}$$

*where we define $\Sigma_{t;h} := \sum_{\tau=1}^{t} X_h(f^\tau)X_h(f^\tau)^T + \lambda \mathbf{I}$ and we assume $\lambda \geq B_X^2$ holds $\forall t \in [T]$.*