# OpenReview forum: "Cal-QL: Calibrated Offline RL Pre-Training for Efficient Online Fine-Tuning"
_ICLR.cc/2023/Workshop/RRL — RRL 2023 Spotlight_

### Official Review · Reviewer_YTsi · 2023-02-21
**simple method with great results and good motivation**

**Rating:** 4
**Confidence:** 4

**Review:**

In this paper, the authors proposed a new method called Calibrated Q-learning and showed that a conservative value function is a necessary but not sufficient condition for fast online fine-tuning. They empirically demonstrated that methods based solely on conservativeness suffer from "unlearning" during fine-tuning, which negates all the advantages of offline initialization, while methods based on implicit constraints (such as IQL) have asymptotically slower learning. To mitigate the "unlearning", the authors proposed a solution based on the observation that learned estimates of conservative values are not calibrated, as consequence samples are wasted to correct the scale of Q-values during fine-tuning. Cal-QL fixes this by making Q-values conservative and calibrated, so that learned value estimates are larger than the ground truth return of policies that are worse than the offline initialization.

The resulting method is simple, has theoretical support, and shows good results on tested benchmarks, significantly outperforming the baselines (especially in terms of cumulative regret). The paper is detailed and clearly written. As a result, I believe this paper is a valuable and highly relevant contribution. I especially like the empirical motivation for the method in section 4.1, which gives the right intuition.

minor comments:
- while in the caption to the Figure 3 mentions that fine-tuning begins only at 50k steps, just from the figure alone, text can be confusing, i.e. text says “Note that the Q-values learned by CQL in the offline phase are much smaller than their ground-truth value as expected”, however starting values are on the same scale as ground-truth and without reading caption very carefully, it is hard to understand that I should look at the 50k on x-axis. I suggest adding visual hint - vertical line to indicate when offline-pretraining is ended.
- “the learned Q-values are larger than the ground-truth return of policies worse than the offline initialization” can be hard to understand for non-native speaker. I suggest to add “of policies which are worse than”.
- I think that claims in the introduction about offline initialization as “less dangerous” approach are not supported in the paper (or in the cited works), as such initialization does not guarantees safety of online fine-tuning in any way.

questions:
- have you tried estimating the $Q^{\mu}(s, a)$ with offline SARSA? It seems to me that such a way would be more accurate and more generalizable than the return-to-go estimation via monte-carlo and regression

---

### Official Review · Reviewer_bpLF · 2023-02-27
**Interesting paper**

**Rating:** 4
**Confidence:** 4

**Review:**

This work proposes to constrain the regularization term of Conservative Q-Learning by enforcing to lower-bound the learned Q value by the (FA-estimated) empirical Q value of the dataset. Such lower-bounded Q values are said to be _calibrated_. This change is motivated by the observation that pretrained Q-values obtained from CQL get very negative, and that when switching to the online phase the Q-values jump back to values closer to the average return. By avoiding these extremely negative values during training, Cal-QL empirically appears to learn faster during the online finetuning phase.

Some thoughts:
- The language around bounding is a bit confusing at times. From what I understand, $Q^\mu$ acts as a **lower-bound** to $Q_\theta$. We want $Q_\theta(s, a)$ to be **at least as high as** $Q^\mu(s,a)$. I'm not sure it's ideal to say that $Q_\theta$ _upper bounds_ $Q_\mu$, because $Q_\theta$ is the quantity of interest. Usually the thing that is _bounded_ is what we are looking for, and it _bounded by_ other (often fixed) quantities. I think it would be easier to understand if the wording was "we lower-bound $Q_\theta$ by $Q^\mu$".
- The word "unlearning" also may not be ideal, and feels like it's doing a lot of work, especially given that the evidence is about (a) the magnitude of Q-values (b) the success rate. During online training, the model is effectively learning about its environment, learning _not_ to take certain actions; that this happens in parameter space and not just in value & performance space matters. I would love to see a deeper analysis of what happens during this phase of low success rate.
- In 4.2, I fail to see why this speculation is correct: "the policy optimizer would not unlearn $\pi$ in favor of a worse $\mu$ upon observing new online data since $\pi$ still attains a larger value under the learned $Q_\theta$ function". The only thing that a calibrated Q value tells us is that _predicted_ action-values are lower-bounded. This does _not_ imply that actual returns are. For both CQL and Cal-QL, the OOD $(s, a)$ are OOD, and the only difference is the prior we put on the solutions the function approximation is allowed to find, which may impact generalization or numerical stability.

Overall the paper is a good workshop contribution. It takes a problem, investigates it, proposes an effective solution and benchmarks it. The writing is good. I think some aspects could be stronger, and things dug deeper, but this is otherwise good work by the simple metric that I learned something new.